# Peak-Return Greedy Slicing: Subtrajectory Selection for Transformer-based Offline RL

**Zhiwei Xu**[1], **Miduo Cui**[1], **Dapeng Li**[3,4], **Zhihao Liu**[2,5], **Haifeng Zhang**[2], **Hangyu Mao**[6], **Guoliang Fan**[2], **Bin Zhang**[2]*

[1] School of Artificial Intelligence, Shandong University, Jinan, China
[2] The Key Laboratory of Cognition and Decision Intelligence for Complex Systems,
  Institute of Automation, Chinese Academy of Sciences, Beijing, China
[3] Li Auto Inc., China
[4] Tsinghua University, Beijing, China
[5] Zhongguancun Academy, Beijing, China
[6] Institute of Microelectronics, Chinese Academy of Sciences, Beijing, China
`zhiwei_xu@sdu.edu.cn`, `zhangbin2020@ia.ac.cn`

## Abstract

Offline reinforcement learning enables policy learning solely from fixed datasets, without costly or risky environment interactions, making it highly valuable for real-world applications. While Transformer-based approaches have recently demonstrated strong sequence modeling capabilities, they typically learn from complete trajectories conditioned on final returns. To mitigate this limitation, we propose the Peak-Return Greedy Slicing (PRGS) framework, which explicitly partitions trajectories at the timestep level and emphasizes high-quality subtrajectories. PRGS first leverages an MMD-based return estimator to characterize the distribution of future returns for state-action pairs, yielding optimistic return estimates. It then performs greedy slicing to extract high-quality subtrajectories for training. During evaluation, an adaptive history truncation mechanism is introduced to align the inference process with the training procedure. Extensive experiments across multiple benchmark datasets indicate that PRGS significantly improves the performance of Transformer-based offline reinforcement learning methods by effectively enhancing their ability to exploit and recombine valuable subtrajectories.

## 1 Introduction

In practice, offline reinforcement learning (RL) (Levine et al., 2020), which learns policies purely from fixed datasets without environment interaction, has garnered significant attention in recent years. Unlike online RL, offline RL eliminates the need for costly or potentially risky interactions with the environment, making it particularly valuable in domains such as autonomous driving (Lin et al., 2024), robotics (Kumar et al., 2023), and recommender systems (Xin et al., 2022). With advances in sequence modeling, Transformer-based offline RL (Chen et al., 2021; Janner et al., 2021) has gained increasing traction. Leveraging the Transformer's strength in modeling long-range dependencies and its expressive power (Vaswani et al., 2017), recent methods have achieved substantial improvements on several benchmark datasets, presenting a novel paradigm for the progress of offline reinforcement learning.

However, existing Transformer-based offline RL methods remain limited in effectively stitching together high-quality segments from different trajectories (Brandfonbrener et al., 2022). The *stitching* capability refers to the algorithm's ability to identify and recombine superior trajectory fragments, thereby enabling policy learning that outperforms what can be achieved from any single existing trajectory. Although some prior works have explored trajectory resampling (Li et al., 2024; Lee et al., 2024), value-based guidance (Yamagata et al., 2023; Pei et al., 2025; Wang et al., 2024), or conditional modeling (Zhang et al., 2024; Kim et al., 2024; Wu et al., 2023) to mitigate this issue,

---

*Corresponding author.

their effectiveness is constrained by the coarse granularity of trajectory-level processing. As a result, these methods often struggle to handle data with uneven quality and fall short of fully leveraging the power of the Transformer.

It is worth noting that humans, when learning to make decisions, do not evaluate an experience solely based on its final outcome. Instead, they tend to distinguish *good subtrajectories* and *bad subtrajectories* within long trajectories. As illustrated in Figure 1, even if a trajectory leads to a suboptimal outcome overall, it may still contain locally high-value experiences. Current Transformer-based methods generally derive insights from complete trajectories, focusing exclusively on their final returns. In contrast, humans selectively retain valuable subtrajectories and enhance learning efficiency by composing new experiences through stitching (Schank & Abelson, 2013; Nakahashi et al., 2016; Dennett, 1989; Schachner & Carey, 2013). This capability of *subtrajectory selection* is notably absent in Transformer-based offline RL methods, primarily because identifying appropriate *split points* within a trajectory is often challenging.

Figure 1: Illustration of trajectory slicing with an appropriate split point.

Inspired by this observation, we propose the **P**eak-**R**eturn **G**reedy **S**licing (**PRGS**) framework to enhance the performance of Transformer-based offline RL methods by enabling them to identify and focus on high-quality experience subtrajectories. Concretely, we first train a return estimator based on Maximum Mean Discrepancy (MMD) (Smola et al., 2006), which approximates the distribution of potential future returns for each state-action pair under *optimistic* assumptions. Leveraging this estimator, PRGS infers the initial-to-go return for each state along a trajectory and recursively segments the entire trajectory into multiple subtrajectories. Positions with *peak return* are identified as *split points*. This fine-grained slicing at the timestep level allows PRGS to effectively preserve valuable experience segments and stitch them. Finally, to ensure consistency in history length between training and evaluation, we introduce an adaptive mechanism that truncates the input history dynamically during evaluation. To evaluate the effectiveness of PRGS, we integrate it into several Transformer-based offline RL algorithms and conduct comprehensive experiments across a variety of benchmark domains. Experimental results demonstrate that PRGS may be seamlessly integrated into existing frameworks and consistently yields an average performance improvement of **15.8%** over the original baselines across diverse tasks.

## 2 PRELIMINARY

### 2.1 OFFLINE REINFORCEMENT LEARNING

In offline RL, an agent is trained solely on a static dataset $\mathcal{D} = \{\tau\}$. Each trajectory $\tau = \{(s, a, r)\}$ consists of state $s \in \mathcal{S}$, action $a \in \mathcal{A}$, and reward $r \in \mathbb{R}$ at each time step. The objective is to learn a policy $\pi : \mathcal{S} \to \mathcal{A}$ that maximizes the expected discounted return $J(\pi) = \mathbb{E}_{\tau \sim \pi}[\sum_{t=0}^{\infty} \gamma^t r_t]$, where $\gamma \in (0, 1]$ is the discount factor. In contrast to online RL (Sutton & Barto, 2018), the key challenge in offline RL is that the policy must be optimized solely based on the historical dataset $\mathcal{D}$, without access to additional interactions with the environment.

### 2.2 TRANSFORMER-BASED OFFLINE RL METHODS

In recent years, the Transformer architecture (Vaswani et al., 2017) has been increasingly applied to offline reinforcement learning, primarily by treating a trajectory $\tau = \{(s_t, a_t, r_t)\}_{t=0}^{K}$ as a sequential input and representing its elements as unified tokens (Janner et al., 2021; Chen et al., 2021; Wu et al., 2023; Wang et al., 2024). Specifically, the state $s_t$, action $a_t$, and return $G_t = \sum_{i=t}^{K} \gamma^i r_i$ are embedded into a shared representation space, thereby enabling the model to capture long-range dependencies within the trajectory. Almost all Transformer-based offline RL methods, such as Transformer-based Behavior Cloning (BC) and Decision Transformer (DT) (Chen et al., 2021), share a unified training objective, which can be formulated as:

$$\mathcal{L}(\theta) = -\mathbb{E}_{\tau \sim \mathcal{D}} \sum_{t=0}^{K} \log \pi_\theta(a_t \mid \tau_{\leq t}), \tag{1}$$

where $\pi_\theta$ denotes a Transformer-parameterized policy conditioned on the trajectory prefix $\tau_{\leq t} = \{\tau_i\}_{i=0}^t$. In Transformer-based BC, the prefix $\tau_{\leq t}$ consists of only $(s_i, a_i)$ pairs. While in DT, it additionally incorporates the return-to-go token $\hat{G}_i$, resulting in $\tau_{\leq t} = \{(G_i, s_i, a_i)\}_{i=0}^t$. $G_t$ mainly serves as a conditioning signal that guides the policy towards trajectories with higher expected returns. For continuous action spaces, the loss in Eq.(1) is equivalent to the mean squared error (Wang & Bovik, 2009), while for discrete action spaces, it corresponds to the cross-entropy loss (Mao et al., 2023). This unified tokenization and sequential modeling paradigm allows Transformer-based offline RL methods to accommodate diverse task formulations within a shared framework and has demonstrated strong modeling performance across various benchmark domains. Unlike traditional offline RL that depends on value-based or policy optimization objectives, Transformer-based methods learn policies purely through sequence modeling.

## 3 METHODOLOGY

In this paper, we propose the Peak-Return Greedy Slicing (PRGS) framework, which comprises three fundamental components: **(I)** a return estimator based on Maximum Mean Discrepancy (MMD) that evaluates the potential return distribution for each state-action pair within a trajectory; **(II)** a subtrajectory slicing strategy that greedily divides trajectories using an optimistically biased initial return estimate, with the resulting subtrajectories used for policy training; and **(III)** an adaptive history truncation mechanism employed during policy evaluation and execution, which dynamically determines whether to retain or discard historical information by comparing changes in the estimated value. The following sections provide a comprehensive description of each component.

### 3.1 MMD-BASED RETURN ESTIMATOR

To assess the potential value of each state-action pair within a trajectory at a finer level of granularity, we introduce an MMD-based return estimator. Maximum Mean Discrepancy (MMD) is a widely used metric for measuring the distance between distributions (Smola et al., 2006). It quantifies the difference between two distributions $X$ and $Y$ in the Reproducing Kernel Hilbert Space (RKHS) (Berlinet & Thomas-Agnan, 2004) and is defined as:

$$\text{MMD}^2(X, Y) = \mathbb{E}_{x, x' \sim X}[k(x, x')] + \mathbb{E}_{y, y' \sim Y}[k(y, y')] - 2\mathbb{E}_{x \sim X, y \sim Y}[k(x, y)], \quad (2)$$

where $k(\cdot, \cdot)$ denotes a kernel function. In practice, MMD enables the estimation of distributional differences using a finite set of sampled data, and has thus been extensively applied in tasks involving distribution fitting and alignment. In RL, value functions are often treated as distributions (Bellemare et al., 2017; Dabney et al., 2018; Jullien et al., 2025), since a single state-action pair can yield diverse future returns. Traditional quantile regression (Jurevckova, 2006) captures this distributional nature by learning a fixed set of quantile points. In contrast, MMD offers a non-parametric alternative for aligning distributions, allowing the return distribution of state-action pairs to be learned directly, rather than merely predicting an expected value. This approach captures uncertainty and the potential variability of state-action pairs more comprehensively, thereby improving the expressiveness and robustness of value estimation.

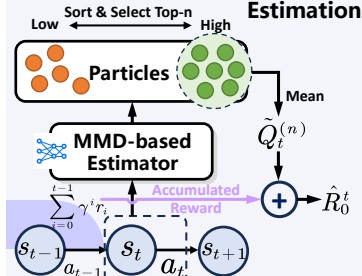

Figure 2: Illustration of MMD-based optimistic return estimation.

Motivated by previous research (Nguyen-Tang et al., 2021), we develop an MMD-based return estimator. Given a *single-step* state $s_t$ and its corresponding action $a_t$, the estimator takes $(s_t, a_t)$ as input and produces a set of $N$ scalar samples:

$$Z_\psi(s_t, a_t) = \{z_1, z_2, \ldots, z_N\}, \quad z_i \in \mathbb{R}, \quad (3)$$

where $\psi$ denotes the model parameters. This set approximates the return distribution conditioned on the given state-action pair, rather than providing a singular estimate. Moreover, unlike estimation methods that condition on entire trajectories, this design reduces the influence of irrelevant historical context. It focuses on evaluating *all possible* intrinsic values of the current state–action pair.

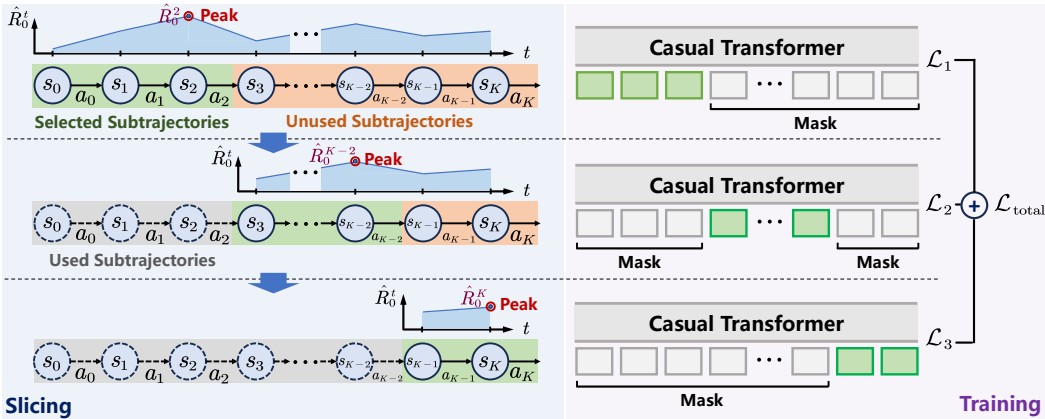

Figure 3: Illustration of the proposed greedy subtrajectory slicing mechanism. Given an offline trajectory, we compute the optimistic return $\hat{R}_0^t$ at each time step and identify the *peak point* $t^*$ as the *split point* for subtrajectory slicing. The selected subtrajectory is used for training while the remaining part is recursively sliced until all time steps are covered.

To train the estimator, we minimize the MMD loss between the predicted distribution $Z_\psi(s_t, a_t)$ and the target distribution $Z^{\text{target}}(s_t, a_t) = \{z_1', z_2', \ldots, z_N'\}$, as shown in Eq.(2):

$$\mathcal{L}_{\text{MMD}} = \frac{1}{N^2} \sum_{i=1}^N \sum_{j=1}^N k(z_i, z_j) + \frac{1}{N^2} \sum_{i=1}^N \sum_{j=1}^N k(z_i', z_j') - \frac{2}{N^2} \sum_{i=1}^N \sum_{j=1}^N k(z_i, z_j'). \tag{4}$$

The target distribution $Z^{\text{target}}$ is computed based on the temporal difference target:

$$Z^{\text{target}}(s_t, a_t) = r_t + \gamma Z_{\psi^-}(s_{t+1}, a_{t+1}),$$

where $\psi^-$ denotes the target network with delayed updates. For simplicity, we adopt a kernel function based on the squared Euclidean distance (Eyster & White, 1973), defined as $k(x, y) = -\|x - y\|^2$. By minimizing the MMD loss above, the estimator is encouraged to learn a stable representation of the value distribution, which serves as a reliable guidance signal for subsequent subtrajectory slicing.

## 3.2 GREEDY SUBTRAJECTORY SLICING

The greedy subtrajectory slicing module constitutes the core of PRGS. For clarity, we provide a detailed exposition of this module by dividing it into three sequential components: *estimation*, *slicing*, and *training*.

**Estimation.** Given an offline trajectory $\tau = \{(s_t, a_t, r_t)\}_{t=0}^K$ of length $K$, we commence by applying the MMD-based return estimator introduced in the previous section to estimate the future return distribution for each state-action pair $(s_t, a_t)$ along the trajectory. To enable consistent comparison across different timesteps, we then derive scalar scores with optimistic bias from the estimated return distributions. According to Eq.(3), the estimator generates $N$ particles at each timestep $t$, representing an approximate distribution of returns starting from $(s_t, a_t)$. To derive an optimistic scalar estimate, the particles are sorted in descending order:

$$z_{t,(1)} \geq z_{t,(2)} \geq \cdots \geq z_{t,(N)}.$$

The value function approximation is then defined as the mean of the top-$n$ particles:

$$\tilde{Q}_t^{(n)}(s_t, a_t) = \frac{1}{n} \sum_{i=1}^n z_{t,(i)}, \tag{5}$$

where $n \in \{1, \ldots, N\}$ is a tunable hyperparameter. Smaller values of $n$ yield estimates closer to the upper quantiles of the distribution, resulting in a more optimistic evaluation, while larger values lead to more conservative estimates. Intuitively, $\tilde{Q}_t^{(n)}$ captures the potential upper-bound

return by focusing on the most favorable outcomes, thereby preventing suboptimal realizations from disproportionately degrading the evaluation of specific subtrajectories.

According to Eq.(C.2), $\tilde{Q}_t^{(n)}$ denotes an optimistic estimate of the return starting from timestep $t$. To ensure comparability across different timesteps, we map this quantity to the perspective of the trajectory's *starting point* $(s_0, a_0)$ by incorporating the cumulative realized reward obtained before $t$, as shown in Figure 2. Formally, we define the aligned optimistic return from the start as:

$$\hat{R}_0^t = \sum_{i=0}^{t-1} \gamma^i r_i + \tilde{Q}_t^{(n)}.$$

In this way, $\hat{R}_0^t$ denotes the optimistic total return the agent would receive by following the observed trajectory up to step $t$ and then executed action $a_t$ at state $s_t$. This alignment brings the potential returns at different timesteps onto a unified scale, enabling consistent comparison and supporting the subsequent greedy segmentation of subtrajectories.

**Slicing.** For the definition of $\hat{R}_0^t$, if there exists a timestep $t^* \in [0, K]$ such that $\hat{R}_0^{t^*} = \max_{0 \leq t \leq K} \hat{R}_0^t$, then the subtrajectory $\tau_{0:t^*} = \{(s_0, a_0, r_0), \ldots, (s_{t^*}, a_{t^*}, r_{t^*})\}$ is identified as the optimal subtrajectory within the trajectory. In other words, learning from this segment is expected to yield a nearly optimal return. Any trajectory segment occurring after the timestep $t^*$ degrades the estimation of $\hat{R}_0^t$, resulting in a suboptimal return value. Based on this observation, we implement a greedy slicing strategy: for each complete trajectory, we select $\tau_{0:t^*}$ corresponding to the peak optimistic return $\hat{R}_0^{t^*}$, and use it as a training sample for the Transformer model. Accordingly, $t^*$ serves as the split point.

**Training.** During training, only the timesteps within $\tau_{0:t^*}$ are retained, while the other steps are masked and excluded from the loss computation. Following Eq.(1), we obtain the modified loss function as:

$$\mathcal{L}_1(\theta) = -\mathbb{E}_{\tau \sim \mathcal{D}} \sum_{t=0}^{t^*} \log \pi_\theta(a_t \mid \tau_{0:t^*}),$$

where $0$ and $t^*$ denote the starting and ending timesteps of the selected subtrajectory, respectively.

Then, the selected subtrajectory $\tau_{0:t^*}$ is labeled as used, and the remaining unused portion $\tau_{t^*+1:K} = \{(s_{t^*+1}, a_{t^*+1}, r_{t^*+1}), \ldots, (s_K, a_K, r_K)\}$ is recursively segmented using the same *greedy strategy*. In the $m$-th recursion step, a new split point is identified to extract a high-quality subtrajectory, and the corresponding loss value $\mathcal{L}_m$ is calculated. This process continues until all timesteps are covered, resulting in a complete decomposition of the original trajectory into a set of disjoint subtrajectories, as illustrated in Figure 3. The overall optimization objective for a training iteration is defined as the weighted sum of the losses across all extracted subtrajectories:

$$\mathcal{L}_{\text{total}} = \sum_{m=1}^{M} \lambda^{m-1} \mathcal{L}_m,$$

where $M$ denotes the number of resulting subtrajectories, which dynamically varies depending on the specific input trajectory. $\lambda \in [0, 1]$ denotes a weighting coefficient, indicating that the contribution of $\mathcal{L}_m$ diminishes as $m$ increases. In particular, when $\lambda = 0$, only the first subtrajectory is considered, whereas $\lambda = 1$ indicates that all subtrajectories are treated uniformly during training. This design aligns with the objective of guiding PRGS to focus more on subtrajectories associated with higher values of $\hat{R}_0$. Meanwhile, this recursive slicing framework ensures that every timestep contributes to training.

## 3.3 ADAPTIVE HISTORY TRUNCATION IN EVALUATION

During the training phase, PRGS applies greedy segmentation to divide the entire trajectory into multiple subtrajectories and trains on each independently. This design makes the initial state of each subtrajectory critical for subsequent decision-making, while the preceding historical context is not explicitly incorporated into training. Consequently, unconditionally retaining all historical information during evaluation introduces a mismatch between training and evaluation, potentially degrading policy performance.

To mitigate this inconsistency, PRGS adopts an adaptive history truncation mechanism during evaluation. At each timestep $t$ ($t \geq 1$), the return estimator provides an optimistic evaluation of the current state, which is then compared with the estimate from the previous step to determine whether the historical trajectory should be retained. First, we define the state value estimate at time step $t$ as:

$$V_t(s_t) = \tilde{Q}_{t-1}^{(n)}(s_{t-1}, a_{t-1}), \quad t \geq 1,$$

where $\tilde{Q}_{t-1}^{(n)}$ denotes an optimistic return based on Eq.(C.2). In the evaluation phase, PRGS compares the estimated value of the current state against that of the preceding step: $\Delta V_t = V_t(s_t) - V_{t-1}(s_{t-1})$. If $\Delta V_t > 0$, it suggests that the current state holds greater potential for return than the previous one, implying that the preceding trajectory no longer contributes valuable information for future decisions. In such cases, PRGS discards the historical trajectory and retains only the current state as the new starting point, from which subsequent decisions are derived. Formally, the history update mechanism during evaluation can be written as:

$$H_t = \begin{cases} \{s_t\}, & \text{if } \Delta V_t > 0, \\ H_{t-1} \cup \{s_t\}, & \text{otherwise,} \end{cases} \tag{6}$$

where $H_t$ denotes the retained trajectory at timestep $t$. This adaptive history truncation strategy ensures consistency between training and evaluation. While training relies on subtrajectories starting from intermediate states, the evaluation phase can also dynamically reset the history based on the return estimator, thereby mitigating the impact of irrelevant or low-quality past information on current decision-making.

In summary, the PRGS framework systematically enhances the ability of Transformer-based offline RL through three tightly integrated modules. The MMD-based return estimator provides optimistic value estimates for single state-action pairs, while greedy slicing identifies and prioritizes high-return subtrajectories for training. During evaluation, adaptive truncation dynamically discards irrelevant history to align inference with training-time segmentation. Together, these components enable timestep-level subtrajectory selection and composition, offering a concise and effective way to improve the stitching ability of Transformer-based offline RL. The implementation details and algorithmic description of PRGS can be found in Appendix A.

## 4 RELATED WORK

### 4.1 OFFLINE REINFORCEMENT LEARNING

Besides the Transformer-based approaches discussed in this paper, several other representative methods have also been developed in offline RL. One prominent line of work involves *policy constraint* methods, such as BCQ (Fujimoto et al., 2019) and BRAC (Wu et al., 2019), which mitigate the bias introduced by out-of-distribution (OOD) (Mao et al., 2024b) actions by constraining the learned policy to remain close to the behavior policy during updates. Another class of methods focuses on *advantage weighting* and *value function regression*. For instance, CQL (Kumar et al., 2020) imposes a conservative penalty on the Q-function to prevent overestimation of OOD actions. At the same time, IQL (Kostrikov et al., 2022) leverages expectile regression to stabilize the learning dynamics. With the development of sequence modeling techniques, researchers have introduced Diffusion Models (Ho et al., 2020) as a more scalable tool for policy representation and planning. Typical representatives include Diffuser (Janner et al., 2022) and DD (Ajay et al., 2023). In addition, EDP (Kang et al., 2023) and DiffuserLite (Dong et al., 2024) optimize the training and inference efficiency of diffusion-based policies. This series of works demonstrates the great potential of diffusion architectures in Offline RL, bringing a new generative model paradigm to offline decision-making.

### 4.2 STITCHING IN TRANSFORMER-BASED OFFLINE RL METHODS

Although Transformer-based approaches have shown strong modeling capabilities in offline RL, they still face limitations in stitching together high-value segments from different parts of trajectories. Several recent works have explored ways to improve this ability. For instance, TT (Janner et al., 2021) incorporates discretization and dynamic programming into trajectory modeling, enabling the model to utilize planning mechanisms for more effective trajectory recombination. Methods such

as RCSL (Srivastava et al., 2019; Schmidhuber, 2019), QDT (Yamagata et al., 2023), CGDT (Wang et al., 2024), and QT (Hu et al., 2024) integrate value estimation into sequential modeling, allowing value-guided composition of trajectory segments. EDT (Wu et al., 2023) estimates the value of past observations and adaptively determines the effective context length based on the estimated utility. Meanwhile, DoC (Yang et al., 2023) takes an alternative approach by learning latent representations of future trajectories and conditioning the policy model on these representations to guide behavior adjustment. And Sun & Wu (2023) propose using reward machines to structure offline data, enabling strong policies to be learned from less but higher-quality data in complex tasks.

The above methods have demonstrated effectiveness in improving trajectory stitching. Nevertheless, they lack a fine-grained perspective to explicitly distinguish between high-quality and low-quality subtrajectories within a trajectory. Most approaches operate at the trajectory level or rely on implicit latent representations, without mechanisms to explicitly identify high-quality subtrajectories at the timestep level. To mitigate this constraint, we present an interpretable mechanism that identifies split points and directly selects informative subtrajectories from a given trajectory. It is compatible with various Transformer-based offline RL algorithms and significantly improves their stitching performance.

## 5 EXPERIMENT

This section empirically evaluates the effectiveness and advantages of PRGS through a series of systematic experiments. The evaluation includes four main components: **(I)** integrating PRGS into several representative Transformer-based offline RL algorithms and assessing its overall performance across several benchmark datasets; **(II)** performing ablation studies to investigate the contributions of individual components and hyperparameters, including the number of particles $n$ in the MMD-based estimator and the adaptive history pruning mechanism; **(III)** verifying the efficacy of segmentation at the timestep level by comparing it with conventional trajectory-level filtering methods; **(IV)** finally, providing visualizations to illustrate the inner workings of PRGS to facilitate understanding of its impact on the training.

### 5.1 OVERALL PERFORMANCE

We conduct experiments on various representative offline RL benchmarks to thoroughly assess the effectiveness of PRGS. The baselines are grouped into three categories: (1) classical offline RL algorithms, including CQL (Kumar et al., 2020), IQL (Kostrikov et al., 2022), BEAR (Kumar et al., 2019), and TD3+BC (Fujimoto & Gu, 2021); (2) recent approaches emphasizing stitching capability, such as QDT (Yamagata et al., 2023), CGDT (Wang et al., 2024), and EDT (Wu et al., 2023); and (3) Transformer-based methods like BC, DT, and PDiT (Mao et al., 2024a). PDiT establishes a framework with enhanced representational capacity while still adhering to Eq.(1), and is therefore adopted as an important baseline in this paper. To demonstrate the effectiveness of the proposed framework, we construct Transformer-based offline RL variants augmented with PRGS, denoted by appending PRGS as a suffix. The evaluation spans diverse domains, including standard continuous control tasks from the D4RL benchmark (Fu et al., 2020), natural language and multi-step reasoning tasks in BabyAI (Chevalier-Boisvert et al., 2019), and auction-style decision-making tasks in AuctionNet (Su et al., 2024). These benchmarks cover both continuous and discrete action spaces, allowing for a more comprehensive comparison. For stitching-focused methods, we restrict the evaluation to scenarios where such capabilities are particularly relevant. In the experimental results, the arrows denote the performance difference between the PRGS algorithm and the baseline algorithm. All experiments are based on publicly available and reproducible implementations, with detailed hyperparameter settings and dataset configurations provided in Appendix A and B. Furthermore, additional experimental results and a more detailed analysis are presented in Appendix C.

The experimental results on the D4RL benchmark (see Table 1) show that the proposed PRGS framework consistently achieves superior performance across a wide range of tasks. For Gym tasks, traditional offline RL algorithms such as CQL and IQL perform reasonably well on *medium* or *expert* datasets, but exhibit a notable drop in performance on *medium-replay* datasets. In contrast, methods enhanced with PRGS demonstrate significantly improved stability and higher average returns. For instance, DT-PRGS shows substantial gains over vanilla DT on *medium-replay* datasets. In the Adroit and Kitchen domains, PRGS excels on several challenging human demonstration datasets (including *pen-human* and *hammer-human*), outperforming existing baselines and demonstrating

Table 1: Offline RL methods on D4RL benchmarks.

| | Scenarios | CQL | IQL | BEAR | TD3+BC | BC | BC-PRGS | DT | DT-PRGS | PDiT | PDiT-PRGS |
|---|---|---|---|---|---|---|---|---|---|---|---|
| **Gym Tasks** | halfcheetah-medium-expert | 91.6 | 86.7 | 53.4 | 90.7 | 86.2±9.4 | 88.8±2.9 | 91.7±0.3 | 93.1±0.4 | 73.0±4.3 | 85.1±2.8 |
| | hopper-medium-expert | 105.4 | 91.5 | 96.3 | 98 | 67.5±13.1 | 71.5±10.0 | 109.8±0.5 | 111.2±0.3 | 111.4±0.1 | 111.3±0.1 |
| | walker2d-medium-expert | 108.8 | 109.6 | 40.1 | 110.1 | 108.7±0.3 | 108.8±0.1 | 108.9±0.1 | 109.6±0.4 | 108.8±0.4 | 106.6±1.6 |
| | halfcheetah-medium | 49.2 | 47.4 | 41.7 | 48.4 | 40.5±0.1 | 40.7±0.3 | 40.0±0.1 | 41.8±0.5 | 42.8±2.3 | 39.9±0.1 |
| | hopper-medium | 69.4 | 66.3 | 52.1 | 59.3 | 59.9±1.5 | 62.4±5.7 | 63.6±2.6 | 93.1±2.7 | 68.2±2.4 | 92.4±2.0 |
| | walker2d-medium | 83.0 | 78.3 | 59.1 | 83.7 | 78.8±1.1 | 77.8±2.6 | 78.1±1.5 | 78.5±0.9 | 77.6±0.6 | 77.9±2.2 |
| | halfcheetah-medium-replay | 45.5 | 44.2 | 38.6 | 44.6 | 35.8±0.7 | 37.9±0.7 | 35.0±1.0 | 39.9±0.3 | 40.8±2.3 | 34.5±0.4 |
| | hopper-medium-replay | 95.0 | 94.7 | 33.7 | 60.9 | 48.0±28.2 | 50.5±23.0 | 78.7±0.3 | 98.1±1.7 | 89.6±2.7 | 95.4±1.6 |
| | walker2d-medium-replay | 77.2 | 73.9 | 19.2 | 81.8 | 57.5±3.3 | 70.8±4.3 | 71.5±1.7 | 81.1±0.8 | 74.1±0.6 | 78.2±4.1 |
| | **Average** | 80.6 | 77.0 | 48.2 | 75.3 | 64.8 | 67.7 ↑2.9 | 75.3 | 82.9 ↑7.6 | 76.3 | 80.1 ↑3.8 |
| **Adroit Tasks** | pen-human | 37.5 | 71.5 | −1.0 | −3.9 | 109.9±7.6 | 118.5±2.8 | 83.6±7.2 | 119.9±6.9 | 97.7±3.9 | 116.6±5.0 |
| | hammer-human | 4.4 | 1.4 | 0.3 | 1.0 | 2.5±0.8 | 7.2±4.3 | 3.0±0.8 | 4.4±2.0 | 3.7±1.7 | 5.8±3.6 |
| | door-human | 9.9 | 4.3 | −0.3 | −0.3 | 14.5±1.9 | 13.5±3.3 | 22.7±4.4 | 7.8±1.6 | 14.1±4.6 | 8.8±5.3 |
| | pen-cloned | 39.2 | 37.3 | 26.5 | 5.1 | 77.8±12.3 | 69.1±5.9 | 60.3±8.7 | 69.3±6.3 | 91.4±14.0 | 80.6±16.0 |
| | hammer-cloned | 2.1 | 2.1 | 0.3 | 0.3 | 2.6±2.4 | 2.0±1.7 | 1.6±0.4 | 4.3±2.2 | 3.8±1.9 | 6.7±0.0 |
| | door-cloned | 0.4 | 1.6 | −0.1 | −0.3 | 1.6±1.0 | 0.3±0.3 | 14.3±1.7 | 5.3±1.8 | 9.5±2.3 | 7.4±0.5 |
| | **Average** | 15.6 | 19.7 | 4.3 | 0.3 | 34.8 | 35.1 ↑0.3 | 30.9 | 35.2 ↑4.3 | 36.7 | 37.7 ↑1.0 |
| **Kitchen Tasks** | kitchen-complete | 43.8 | 62.5 | 0.0 | 0.0 | 48.3±3.8 | 70.8±2.9 | 40.3±5.3 | 71.7±2.9 | 27.5±0.2 | 31.7±6.3 |
| | kitchen-partial | 49.8 | 46.3 | 13.1 | 0.0 | 52.5±12.5 | 50.0±4.3 | 59.9±4.2 | 59.2±8.0 | 43.3±8.0 | 50.8±2.9 |
| | **Average** | 46.8 | 54.4 | 6.6 | 0.0 | 50.4 | 60.4 ↑10.0 | 50.1 | 65.5 ↑15.4 | 35.4 | 41.3 ↑5.9 |
| **Maze2D Tasks** | maze2d-umaze | 94.7 | 42.1 | 65.7 | 14.8 | 16.4±4.7 | 33.9±11.5 | 60.3±7.7 | 82.4±1.1 | 73.2±11.6 | 82.3±5.0 |
| | maze2d-medium | 41.8 | 34.9 | 25.0 | 62.1 | 20.1±8.4 | 19.3±1.8 | 37.0±8.3 | 90.4±6.8 | 51.2±4.9 | 86.5±15.2 |
| | maze2d-large | 49.6 | 61.7 | 81.0 | 88.6 | 10.3±8.6 | 9.3±1.7 | 25.4±4.9 | 127.5±36.2 | 40.0±10.2 | 106.6±20.4 |
| | **Average** | 62.0 | 46.2 | 57.2 | 55.2 | 15.6 | 20.8 ↑5.2 | 40.9 | 100.1 ↑59.2 | 54.8 | 91.8 ↑37.0 |
| **AntMaze Tasks** | antmaze-umaze | 74.0 | 87.5 | 73.0 | 78.6 | 82.3±3.2 | 83.7±10.0 | 67.0±1.7 | 96.3±1.5 | 89.3±4.9 | 71.0±13.0 |
| | antmaze-umaze-diverse | 84.0 | 62.2 | 61.0 | 71.4 | 81.7±9.5 | 83.0±1.0 | 63.0±4.6 | 80.3±3.8 | 60.3±6.1 | 66.3±9.0 |
| | antmaze-medium-diverse | 53.7 | 70.0 | 8.0 | 3.0 | 2.0±1.0 | 2.0±1.0 | 2.7±1.2 | 1.3±0.6 | 3.3±5.8 | 0.0±0.0 |
| | antmaze-large-diverse | 14.9 | 47.5 | 0.0 | 0.0 | 0.0±0.0 | 0.0±0.0 | 1.0±1.0 | 16.7±7.8 | 1.7±1.5 | 11.0±3.0 |
| | **Average** | 56.7 | 66.8 | 57.2 | 38.3 | 41.5 | 42.2 ↑0.7 | 33.4 | 48.7 ↑15.3 | 38.7 | 37.1 ↓1.6 |

strong robustness to sparse rewards and noisy trajectories. On Maze2D and AntMaze tasks, which involve long-horizon planning and trajectory stitching, PRGS again shows effectiveness. Notably, DT-PRGS achieves a score of 127.5 on *maze2d-large*, outperforming all compared methods. Overall, the PRGS family consistently matches or exceeds the best-performing baselines across diverse settings. These findings validate the effectiveness of the proposed explicit subtrajectory segmentation mechanism in improving the performance of Transformer-based offline RL.

The results on Gym *medium* and *medium-replay* tasks are reported in Table 2. Overall, the DT-based PRGS method consistently outperforms other variants. In particular, on the *hopper-medium-replay* and *walker2d-medium-replay* tasks, PRGS achieves significantly better performance than CGDT, demonstrating its ability to better capture critical subtrajectories in environments with complex dynamics and long-term dependencies. On average, PRGS ranks first among the five methods, achieving a 10.9-point improvement over vanilla DT. These results indicate that PRGS effectively enhances performance, especially excelling in scenarios where replay data is noisy and cross-trajectory recomposition is required.

To highlight the practical utility of the proposed algorithm, we further evaluate various baselines and their PRGS-enhanced variants on the AuctionNet benchmark. As shown in Table 3, PRGS consistently improves the performance scores of the original algorithms in the advertising bidding scenario, with particularly notable gains for BC. This may be attributed to the relatively simple mechanism of BC, which offers greater potential for performance enhancement on AuctionNet.

Finally, in the BabyAI benchmark, we evaluate multiple tasks of varying difficulties. The PDiT method, which employs a multi-layer Transformer architecture, achieves strong performance in this domain. Table 4 reports the results of several Transformer-based offline RL methods alongside their PRGS variants. Across almost all tasks, the PRGS variants consistently outperform the original algorithms, with particularly notable gains on the more challenging tasks.

## 5.2 ABLATION STUDIES

To further disentangle the contribution of each proposed mechanism, extensive ablation studies were conducted. The particle number $n$ was first examined to assess the influence of optimistically biased

Table 2: Results on Gym *medium* and *medium-replay* tasks.

| Gym Tasks | DT | QDT | EDT | CGDT | PRGS |
|---|---|---|---|---|---|
| halfcheetah-medium | 40.0 | 42.2 | 42.5 | 43.0 | 41.8±0.5 |
| hopper-medium | 63.6 | 65.3 | 63.5 | 96.9 | 93.1±2.7 |
| walker2d-medium | 78.1 | 70.1 | 72.8 | 79.1 | 78.5±0.9 |
| halfcheetah-medium-replay | 35.0 | 35.7 | 37.8 | 40.4 | 39.9±0.3 |
| hopper-medium-replay | 78.7 | 55.3 | 89.0 | 93.4 | 98.1±1.7 |
| walker2d-medium-replay | 71.5 | 59.1 | 74.8 | 78.1 | 81.1±0.8 |
| **Average** | 61.2 | 54.6 | 63.4 | 71.8 | **72.1** ↑10.9 |

Table 3: Performance comparison in different periods on AuctionNet.

| Period | BC | BC-PRGS | DT | DT-PRGS | PDiT | PDiT-PRGS |
|---|---|---|---|---|---|---|
| P14 | 284.9±31.4 | 300.1±12.9 | 282.0±14.4 | 297.0±15.5 | 301.4±11.0 | 304.1±9.0 |
| P15 | 296.8±17.4 | 306.6±13.9 | 290.9±11.2 | 293.4±21.1 | 285.1±14.7 | 294.2±10.8 |
| P16 | 265.6±21.2 | 274.5±13.8 | 270.2±8.9 | 278.5±12.9 | 254.4±27.1 | 265.2±17.1 |
| P17 | 294.9±29.6 | 302.6±17.6 | 285.3±14.2 | 303.2±22.5 | 303.8±10.6 | 294.7±16.5 |
| P18 | 219.3±19.9 | 231.2±20.4 | 216.5±11.1 | 225.4±19.0 | 233.4±19.2 | 236.6±15.2 |
| P19 | 286.4±16.0 | 292.2±16.2 | 285.4±13.5 | 284.9±12.7 | 285.3±13.7 | 283.2±12.7 |
| P20 | 241.6±13.6 | 245.7±8.4 | 241.0±9.9 | 232.5±11.6 | 239.7±18.8 | 241.2±11.5 |
| **Average** | 269.9 | **279.0** ↑9.1 | 267.3 | **273.6** ↑6.3 | 271.9 | **274.2** ↑2.3 |

Table 4: Results on BabyAI tasks.

| Scenarios | BC | BC-PRGS | DT | DT-PRGS | PDiT | PDiT-PRGS |
|---|---|---|---|---|---|---|
| GoToRedBall | 87.9±1.7 | 89.6±5.3 | 90.7±2.2 | 91.6±1.6 | 99.2±0.1 | 98.9±0.1 |
| GoToLocal | 77.2±0.1 | 83.0±6.0 | 81.4±1.7 | 82.5±3.6 | 94.4±1.8 | 95.1±0.4 |
| GoToSeq | 34.4±4.0 | 38.7±1.3 | 38.8±1.8 | 39.4±1.0 | 35.5±1.1 | 38.7±3.4 |
| PutNextLocal | 17.5±3.6 | 19.0±1.7 | 13.1±1.6 | 17.1±4.7 | 29.9±2.9 | 32.7±5.1 |
| UnlockLocalDist | 59.1±3.6 | 59.5±3.6 | 59.7±1.9 | 61.8±0.7 | 72.3±4.7 | 81.5±7.0 |
| BossLevel | 36.3±3.2 | 42.2±2.9 | 42.0±1.4 | 43.2±5.3 | 46.4±4.3 | 48.3±3.8 |
| **Average** | 52.1 | **55.3** ↑3.2 | 54.3 | **55.9** ↑1.6 | 63.0 | **65.9** ↑2.9 |

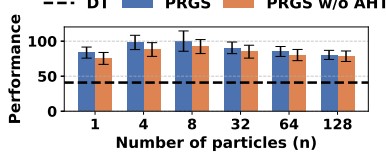

Figure 4: Ablation studies.

estimates, and the necessity of the adaptive history truncation mechanism as an alignment strategy during evaluation was also investigated.

Experiments on the Maze2D dataset provide the ablation results summarized in Figure 4. When the particle number $n$ is very small, the results are suboptimal, likely because the MMD-based estimator generates outlier estimates that impair accuracy. When $n$ is excessively large, the method degenerates into a standard value estimator, losing its optimistic bias and resulting in degraded performance. On the other hand, the variant denoted as PRGS w/o AHT, which excludes the adaptive history truncation mechanism, shows a clear performance drop, indicating that the adaptive history truncation mechanism plays a crucial role in maintaining alignment between training and evaluation. Additional ablation results about the coefficient $\lambda$ are reported in Appendix C.

## 5.3 EFFECTIVENESS OF TIMESTEP LEVEL SUBTRAJECTORY SLICING

To validate that the performance improvement achieved by PRGS arises from its explicit timestep-level subdivision rather than merely filtering trajectories for training, we evaluate several PRGS variants and compare them with results obtained from training on Maze2D datasets filtered to retain only a small fraction of high-quality trajectories.

Experimental results in Figure 5 demonstrate that timestep-level subtrajectory slicing significantly enhances the performance of various baseline methods. In particular, models trained on the top 10% or 20% of trajectories do not necessarily outperform the original algorithms, a phenomenon especially evident in BC. By contrast, PRGS consistently surpasses other slicing strategies across all three base models, confirming the effectiveness and generality of timestep-level

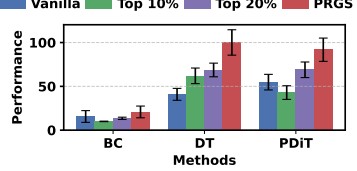

Figure 5: Results of different trajectory selection strategies.

subtrajectory slicing in improving policy learning. These findings suggest that finer-grained subtrajectory selection enables more stable and efficient training across diverse offline RL frameworks.

## 5.4 VISUALIZATION

In this section, we present a visual analysis of trajectory data in *maze2d-medium*. Figure 6(a) shows that the raw trajectories span most of the maze, yet the majority fail to reach or approach the target (marked by a red star). In Figure 6(b), trajectory points are color-coded by their return-to-go (Rtg), with brighter colors corresponding to higher cumulative returns; this highlights that high-return regions are concentrated around the target. Figure 6(c) illustrates the estimates of $\tilde{Q}$ for state–action pairs obtained from the MMD-based estimator. While the overall trend aligns with the distribution of return-to-go, it additionally distinguishes value variations in distant regions, suggesting that the method captures local values more effectively and yields more optimistic estimates. Finally, Figure 6(d) depicts the *first* subtrajectory selected by PRGS during the slicing process. PRGS consistently favors subtrajectories that include high-value regions while discarding low-quality segments

that drift away from the target, thereby showing its ability to automatically extract the most informative portions of complex trajectory sets for learning.

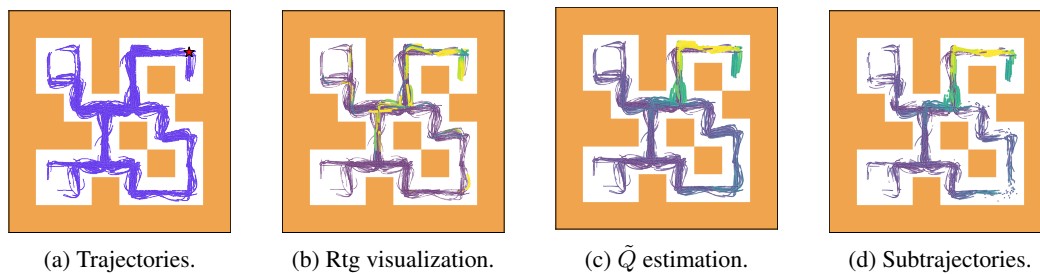

(a) Trajectories.      (b) Rtg visualization.      (c) $\tilde{Q}$ estimation.      (d) Subtrajectories.

Figure 6: Visualization of trajectory data in *maze2d-medium*.

## 6    CONCLUSION

This paper introduces Peak-Return Greedy Slicing (PRGS), which integrates an MMD-based return estimator, a greedy subtrajectory slicing strategy, and an adaptive history truncation mechanism to explicitly select high-quality subtrajectories at the timestep level. This approach substantially improves the compositional capability of Transformer-based offline RL. Experimental results on multiple benchmark tasks validate the effectiveness of PRGS, highlighting its potential as a concise and efficient framework and suggesting promising directions for extensions to more complex environments. And extending this work to non-transformer-based methods is a interesting direction.

## REPRODUCIBILITY STATEMENT

We make every effort to ensure the reproducibility of our work. The paper provides a complete description of the proposed PRGS framework in Section 3, including algorithmic details and training objectives. All experimental settings, benchmark descriptions, and evaluation metrics are reported in Section 5, with additional hyperparameter details and implementation notes included in Appendix A. The datasets employed (D4RL, AuctionNet, BabyAI) are publicly accessible, and we describe the preprocessing steps in Appendix B. An link to the source code is provided at https://github.com/deligentfool/PRGS, and the code is also available in the supplementary material together with instructions for running the experiments.

## ACKNOWLEDGMENTS

This work was supported by the National Natural Science Foundation of China (Grant No.62506210).

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

# A  IMPLEMENTATION DETAILS

## A.1  ALGORITHMIC DESCRIPTION

The algorithm for PRGS are summarized in Algorithm 1. The code for PRGS can be found in the supplementary material.

---

**Algorithm 1:** Peak-Return Greedy Slicing (PRGS)

---

**Input:** Offline dataset $\mathcal{D} = \{\tau\}$; Return estimator $Z_\psi$; Target estimator $Z_{\psi-}$;
Transformer-based policy $\pi_\theta$; Discount factor $\gamma$; Number of particles $N$; Number of
Chosen top particles $n$; Coefficient $\lambda$

1  **Training Phase:**
2  **for** *each trajectory* $\tau = \{(s_t, a_t, r_t)\}_{t=0}^K \in \mathcal{D}$ **do**
3  $\quad$ Sample particles $Z$ and target particles $Z^{\text{target}}$
4  $\quad$ Compute MMD loss and update parameters $\psi$ of the estimator
5  $\quad$ Periodically update target estimator $Z_{\psi-} \leftarrow Z_\psi$
6  $\quad$ Initialize unused timestep set $\mathcal{U} \leftarrow \{0, \dots, K\}$
7  $\quad$ **while** $\mathcal{U}$ *is not empty* **do**
8  $\quad\quad$ $\mathcal{L}_{\text{total}} \leftarrow 0$
9  $\quad\quad$ $t_{\text{prev}}^\star \leftarrow 0$
10 $\quad\quad$ $m \leftarrow 1$
11 $\quad\quad$ **for** $t \in \mathcal{U}$ **do**
12 $\quad\quad\quad$ Obtain $N$ particles: $Z_\psi(s_t, a_t) = \{z_{t,1}, \dots, z_{t,N}\}$
13 $\quad\quad\quad$ Sort in descending order to get $z_{t,(1)} \geq \cdots \geq z_{t,(N)}$
14 $\quad\quad\quad$ Compute optimistic value: $\tilde{Q}_t^{(n)} \leftarrow \frac{1}{n} \sum_{i=1}^n z_{t,(i)}$
15 $\quad\quad\quad$ Compute aligned optimistic return: $\hat{R}_0^t \leftarrow \sum_{k=0}^{t-1} \gamma^k r_k + \tilde{Q}_t^{(n)}$
16 $\quad\quad$ $t^\star \leftarrow \arg\max_{t \in \mathcal{U}} \hat{R}_0^t$
17 $\quad\quad$ Select subtrajectory $\tau_{t_{\text{prev}}^\star : t^\star}$ and mask all other steps
18 $\quad\quad$ Compute loss $\mathcal{L}_m$ via a masked Transformer forward pass
19 $\quad\quad$ Mark $\{t_{\text{prev}}^\star, \dots, t^\star\}$ as used and set $\mathcal{U} \leftarrow \mathcal{U} \setminus \{0, \dots, t^\star\}$
20 $\quad\quad$ $t_{\text{prev}}^\star \leftarrow t^\star + 1$
21 $\quad\quad$ $\mathcal{L}_{\text{total}} \leftarrow \mathcal{L}_{\text{total}} + \lambda^{m-1} \mathcal{L}_m$
22 $\quad\quad$ $m \leftarrow m + 1$
23 $\quad$ Update the parameters $\theta$ of the policy

24 **Evaluation Phase:**
25 **for** *each rollout episode* **do**
26 $\quad$ Initialize history $H \leftarrow \emptyset$
27 $\quad$ **for** *each step* $t$ **do**
28 $\quad\quad$ **if** $t > 0$ **then**
29 $\quad\quad\quad$ Estimate value $V_t(s_t) \leftarrow \tilde{Q}_t^{(n)}(s_{t-1}, a_{t-1})$
30 $\quad\quad\quad$ **if** $V_t(s_t) > V_{t-1}(s_{t-1})$ **then**
31 $\quad\quad\quad\quad$ Discard history: $H \leftarrow \{(s_t, a_t)\}$
32 $\quad\quad$ Predict next action $a_t \sim \pi_\theta(H)$
33 $\quad\quad$ Append $(s_t, a_t)$ to $H$

---

## A.2  HYPERPARAMETERS

Hyperparameters were based on the implementation of Decision Transformer and are listed in Table 5. Most shared hyperparameters were used without extensive tuning. All experiments in this paper are run on Nvidia GeForce RTX 3090 graphics cards and Intel(R) Xeon(R) Platinum 8280 CPU.

Table 5: Hyperparameters used in experiments.

| Hyperparameter | Value |
| --- | --- |
| Number of layers | 3 |
| Number of attention heads | 1 |
| Embedding dimension | 128 |
| Activation function | ReLU |
| Positional encoding | No for D4RL and BabyAI, Yes for AuctionNet |
| Batch size | 64 |
| Context length $K$ | 20 |
| Dropout | 0.1 |
| Learning rate | 1e-4 |
| Grad norm clip | 0.25 |
| Weight decay | 1e-4 |
| Learning rate warmup | linear warmup for 10000 training steps |
| Training epochs | 10 |
| Eval episodes | 32 |
| Discount factor $\gamma$ | 1 |
| Number of particles $N$ | 128 |
| Number of Chosen top particles $n$ | 8 |
| Coefficient $\lambda$ | 0.7 |
| Target update period for MMD-based estimator | every 50 training steps |

## A.3 NOTIONS

For clarity and to ensure consistent notation, this paper adopts a set of symbols throughout the training and evaluation processes. The primary notations and their definitions are presented in Table 6.

Table 6: Summary of the main notations.

| Notation | Description |
| --- | --- |
| $\mathcal{D}$ | Offline dataset containing trajectories $\tau$. |
| $\tau = \{(s_t, a_t, r_t)\}_{t=0}^{K}$ | A trajectory with states, actions, and rewards of length $K$. |
| $s_t, a_t, r_t$ | State, action, and reward at time step $t$. |
| $Z_\psi(s_t, a_t)$ | Return estimator output for state–action pair $(s_t, a_t)$, represented as $N$ particles. |
| $\psi$ | Parameters of the return estimator $Z_\psi$. |
| $\psi^-$ | Parameters of the target return estimator $Z_{\psi^-}$. |
| $z_{t,(i)}$ | The $i$-th sampled particle from $Z_\psi(s_t, a_t)$. |
| $G_t$ | Return-to-go at step $t$. |
| $\tilde{Q}_t^{(n)}$ | Optimistic value of $(s_t, a_t)$ computed by averaging top-$n$ particles. |
| $\hat{R}_0^t$ | Aligned optimistic return from the trajectory starting point to step $t$. |
| $t^\star$ | The peak step where $\hat{R}_0^t$ achieves its maximum. |
| $\tau_{t_1:t_2}$ | Subtrajectory sliced from step $t_1$ to $t_2$. |
| $\mathcal{U}$ | Set of unused timesteps during greedy slicing. |
| $\pi_\theta$ | Transformer-based policy parameterized by $\theta$. |
| $\mathcal{L}_{\text{MMD}}$ | MMD loss for training the return estimator. |
| $\mathcal{L}_m$ | Training loss computed from the $m$-th selected subtrajectory. |
| $\mathcal{L}_{\text{total}}$ | Total loss aggregated from all subtrajectory losses. |
| $\gamma$ | Discount factor in Bellman updates. |
| $N$ | Number of particles sampled by return estimator. |
| $n$ | Number of top particles used for optimistic return computation. |
| $\lambda$ | Coefficient for weighting successive subtrajectory losses. |
| $H$ | Retained history during evaluation. |
| $V_t(s_t)$ | Estimated optimistic value of current state at time step $t$. |

# B EXPERIMENT DETAILS

## B.1 INTRODUCTION FOR BENCHMARKS

This study evaluates the proposed method on multiple standard benchmark tasks, covering continuous control, auction bidding, and instruction-driven navigation and manipulation. Screenshots of all tasks are provided in Figure 7.

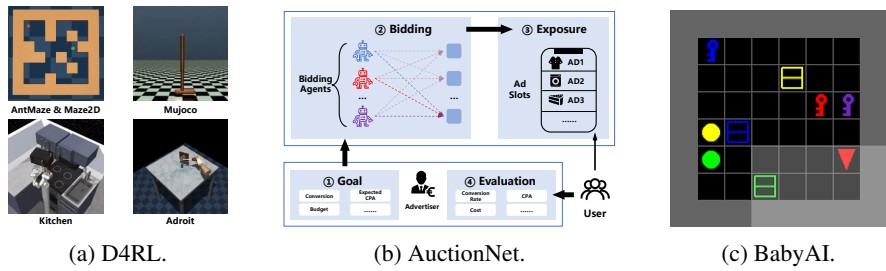

(a) D4RL.      (b) AuctionNet.      (c) BabyAI.

Figure 7: Screenshots of different benchmarks.

**D4RL**. D4RL (Datasets for Deep Data-Driven Reinforcement Learning) is a widely used benchmark suite for offline reinforcement learning. It includes a variety of continuous control environments, such as MuJoCo and Adroit control tasks (*hopper*, *walker2d*, *halfcheetah*, etc.), AntMaze and Maze2D navigation tasks, as well as Kitchen compositional manipulation tasks. A key feature of D4RL is its provision of diverse datasets collected from expert, random, and medium-performance policies, enabling comprehensive evaluation of offline reinforcement learning algorithms under varying data quality conditions.

**AuctionNet**. AuctionNet simulates online advertising bidding, with data derived from large-scale auction logs. The bidding process is formulated as a reinforcement learning problem, where the agent's actions correspond to bid decisions, and the reward function is defined in terms of conversion rates and budget constraints. AuctionNet is commonly employed to investigate policy learning under limited budgets, providing a testbed for assessing the adaptability and robustness of offline reinforcement learning algorithms in real-world industrial applications.

**BabyAI**. BabyAI is an interactive grid-world environment designed to evaluate agents in navigation and manipulation tasks driven by natural language instructions. The environment is partially observable and provides a wide range of subtasks (e.g., picking up, placing, opening doors), which can be composed into more complex tasks. Its trajectory data includes state observations, actions, and language instructions, supporting research on the generalization ability of reinforcement learning and sequence modeling methods conditioned on instructions.

## B.2 EXPERIMENTAL SETUP

In the D4RL environment, we employ the normalized score as the primary evaluation metric to enable direct comparison with prior work. For AuctionNet, the evaluation focuses on cumulative conversion profit and budget compliance. We utilize the Alibaba simulation dataset from the NeurIPS 2024 competition (Xu et al., 2024), spanning 21 periods (7–27), each comprising over 500,000 impression opportunities and 48 auction steps, with each impression involving 48 bidding agents. The dataset contains more than 500 million records, including predicted conversion values, bid information, auction details, and impression outcomes. To ensure fair evaluation, data from periods 7–13 are used for training, while periods 14–20 are reserved for testing. For BabyAI, we adopt success rate as the evaluation metric to assess the agent's ability to execute tasks under natural language instructions. The offline dataset is constructed using heuristic sampling, with 2000 trajectories generated for each scenario to evaluate instruction execution and generalization capability. All reported results are averaged over three independent runs with standard deviations to mitigate the randomness introduced by different seeds.

# C ADDITIONAL RESULTS

## C.1 ADDITIONAL VISUALIZATION

The additional visualizations in Maze2D are shown in Figure 8 and Figure 9. In the *maze-large* environment, the original dataset comprises numerous trajectories spanning the entire map, with a substantial proportion failing to reach the goal. This leads to a mixture of high- and low-quality segments. Visualization of the return-to-go reveals that only trajectories approaching the goal exhibit high returns, whereas those farther away predominantly correspond to low-return regions. The MMD-based value estimator produces results that align with the overall trend of return-to-go but further distinguishes local value variations even in trajectories that do not directly reach the goal, thereby offering more fine-grained value signals. Leveraging these estimations, the PRGS sub-trajectory slicing mechanism prioritizes segments containing high-value regions, effectively discarding uninformative trajectories from exploration and substantially improving the quality of training data.

By contrast, the *maze-umaze* environment has a simpler structure, where trajectories are concentrated in the U-shaped corridor, and most of them cover areas near the goal, yielding overall higher-quality data. Visualizations of return-to-go and the estimator demonstrate a clearer return gradient in this environment, with estimated values strongly correlated with spatial positions. In such cases, although PRGS subtrajectory slicing introduces less dramatic changes than in *maze-large*, it still ensures that training emphasizes critical high-value segments, thereby reinforcing the model's exploitation of key states while maintaining stability.

This comparison demonstrates that PRGS provides powerful filtering and segment selection in complex environments, while in simpler environments it functions as a stable value-alignment mechanism.

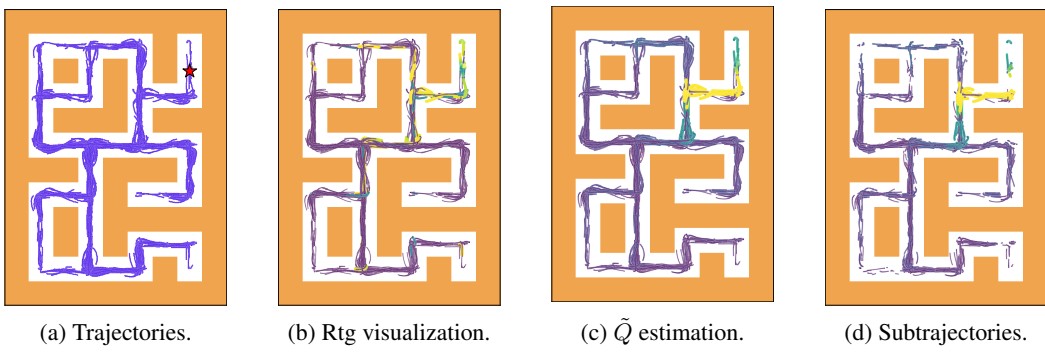

(a) Trajectories.  (b) Rtg visualization.  (c) $\tilde{Q}$ estimation.  (d) Subtrajectories.

Figure 8: Visualization of trajectory data in *maze-large*.

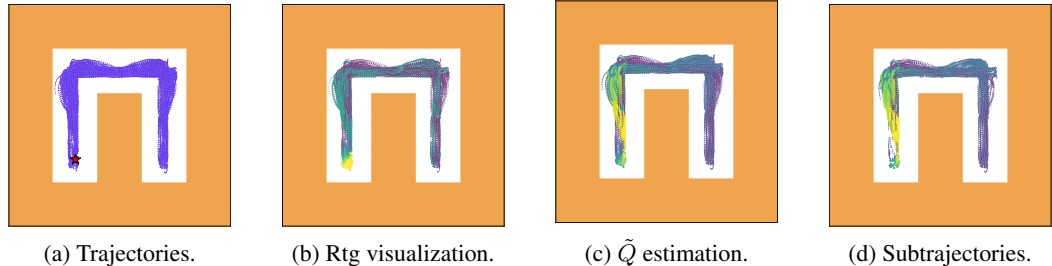

(a) Trajectories.  (b) Rtg visualization.  (c) $\tilde{Q}$ estimation.  (d) Subtrajectories.

Figure 9: Visualization of trajectory data in *maze-umaze*.

Within the *hopper-medium* environment, we perform t-SNE (van der Maaten & Hinton, 2008) embedding and visualization of trajectory data to compare the distributions of return-to-go $G$ and the MMD-based $\tilde{Q}$ estimates. As illustrated in Figure 10(a), the visualization of return-to-go reveals the diversity of trajectory quality in the dataset, where the point colors vary continuously from low

values (purple) to high values (yellow), indicating a substantial mixture of high- and low-return samples within the same embedding space. In Figure 10(b), the $\tilde{Q}$-based estimates follow a similar overall trend to return-to-go but delineate local structures more clearly: certain boundary regions are estimated as high-value, whereas these regions are less evident in the return-to-go visualization.

This suggests that $\tilde{Q}$ can recognize and emphasize potential high-quality state–action pairs in the modeling process, even when their actual returns in the raw data are obscured by noise or suboptimal trajectories.

Overall, these findings demonstrate that the MMD-based return estimator in the *hopper-medium* task not only captures the global value trends of trajectories but also identifies local value differences at a finer granularity, thereby providing more discriminative criteria for subsequent subtrajectory slicing and high-quality experience selection.

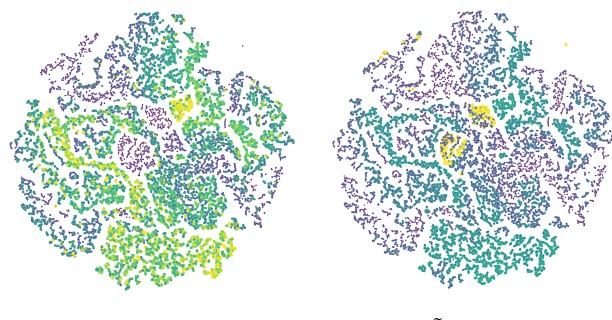

(a) Rtg visualization.      (b) $\tilde{Q}$ estimation.

Figure 10: Visualization of t-SNE embedding trajectory data in *hopper-medium*. Compared to the Rtg visualization, the $\tilde{Q}$ estimation highlights more pronounced color differences in the middle region.

## C.2 ADDITIONAL ABLATION STUDY

To investigate the influence of $\lambda$, which controls the weighting of successive subtrajectories, we conducted experiments in *hopper-medium* and *maze2d-large*, as shown in Figure 11. The results indicate that the optimal value of $\lambda$ varies across different scenarios and algorithms, highlighting its sensitivity to task characteristics. For example, in *hopper-medium* (Figure 11(a)), $\lambda = 0.7$ yields the best performance for DT; while for BC, $\lambda = 0.3$ leads to the most significant improvement. Nevertheless, to better demonstrate the robustness of PRGS, we did not perform task-specific fine-tuning of $\lambda$, but instead adopted a unified setting throughout all experiments. It is also noteworthy that when $\lambda = 0$, meaning that only the first sliced subtrajectory is considered, the method still achieves competitive performance, especially in *maze2d-large*. This observation further validates the key insight that explicit slicing of trajectories at the timestep level is both meaningful and beneficial.

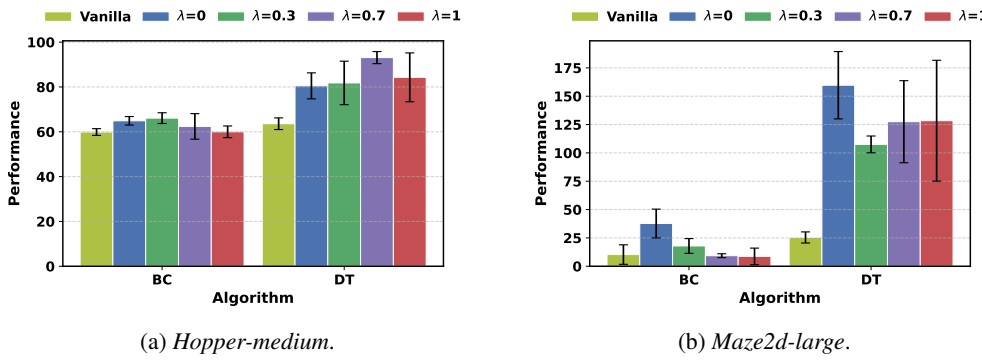

(a) *Hopper-medium*.          (b) *Maze2d-large*.

Figure 11: Influence of the $\lambda$ for PRGS.

We additionally conduct ablations focusing on the MMD-based return estimator. As shown in Figure 12, the full PRGS consistently achieves the best performance on both *hopper* tasks, whereas removing the optimistic MMD estimator (w/o Optimistic) or replacing it with conventional Q-learning (w/ Q-loss) leads to substantial performance drops. Without the top-$n$ particle selection, the return estimates become overly conservative, weakening the ability to identify high-quality trajectory segments; replacing the

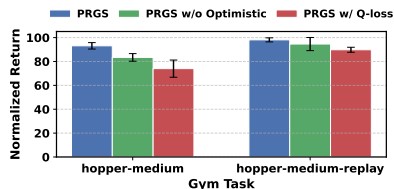

Figure 12: Ablation of the MMD return estimator.

estimator with Q-learning further amplifies offline Q-value bias and significantly harms sequence modeling stability. These results demonstrate that the optimistic MMD return estimator is an essential component of PRGS, enabling more reliable identification of high-value behavioral patterns and ultimately improving slicing quality and overall performance.

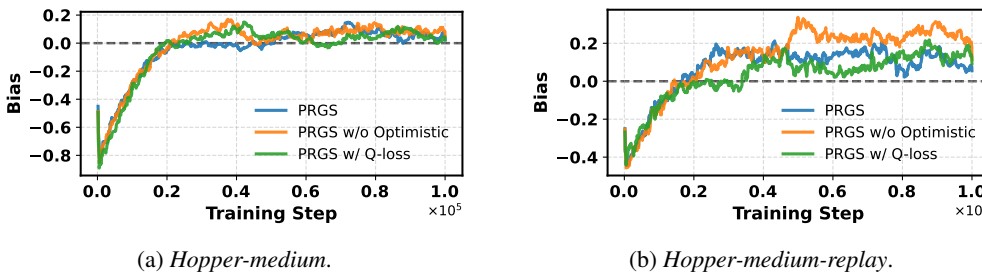

(a) *Hopper-medium.*            (b) *Hopper-medium-replay.*

Figure 13: Bias (related to overestimation) comparison.

To evaluate whether the optimistic MMD-based return estimator leads to significant overestimation issues, we tested above three variants. The bias is calculated as the difference between the estimated return and the true return:

$$\text{Bias} = \mathbb{E}[\tilde{Q}_t - G_t],$$

where $\tilde{Q}_t$ is the estimated return and $G_t$ is the true return. As shown in Figure 13, the PRGS method does not exhibit noticeable overestimation and maintains a stable bias throughout training, compared to the other two variants. Empirically, a properly chosen target update period for the MMD-based estimator keeps the distributional estimates stable.

## C.3 ADDITIONAL ANALYSIS

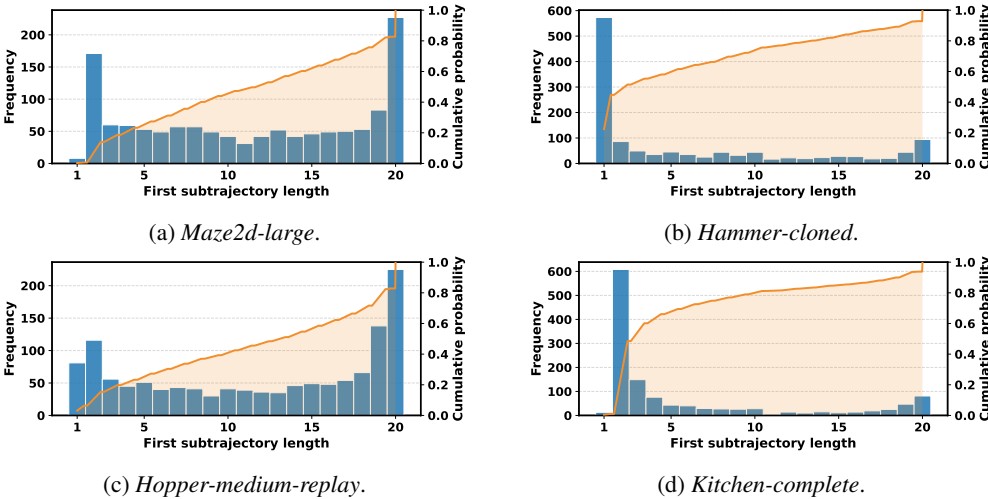

(a) *Maze2d-large.*            (b) *Hammer-cloned.*

(c) *Hopper-medium-replay.*            (d) *Kitchen-complete.*

Figure 14: Distributions of *first* subtrajectory lengths across different environments. The histograms (blue bars, left axis) show the frequency of the first subtrajectory lengths, while the orange curves (right axis) depict the smoothed empirical cumulative distribution functions (ECDFs).

Figure 14 reports the empirical distribution of the length of the *first* subtrajectory selected by PRGS across different benchmark environments. For each environment, about 2000 trajectories of length 20 are randomly sampled. The blue histograms denote the frequency of occurrence, while the orange curves represent the empirical cumulative distribution function (ECDF). The results show that in most environments the first subtrajectory length is concentrated in a relatively short range. For instance, in *hammer-cloned* and *kitchen-complete* the proportion of length-1 subtrajectories dominates, whereas *maze2d-large* and *hopper-medium-replay* exhibit a more dispersed distribution.

Overall, short subtrajectories account for a large proportion across tasks, indicating that the initial slicing step tends to yield short segments. The variation across environments can be attributed to the inherent trajectory characteristics: in tasks with frequent high-reward signals or strong local cues, the first peak is often reached quickly, while in long-horizon navigation or locomotion tasks the return peaks tend to appear later, leading to longer subtrajectory slices.

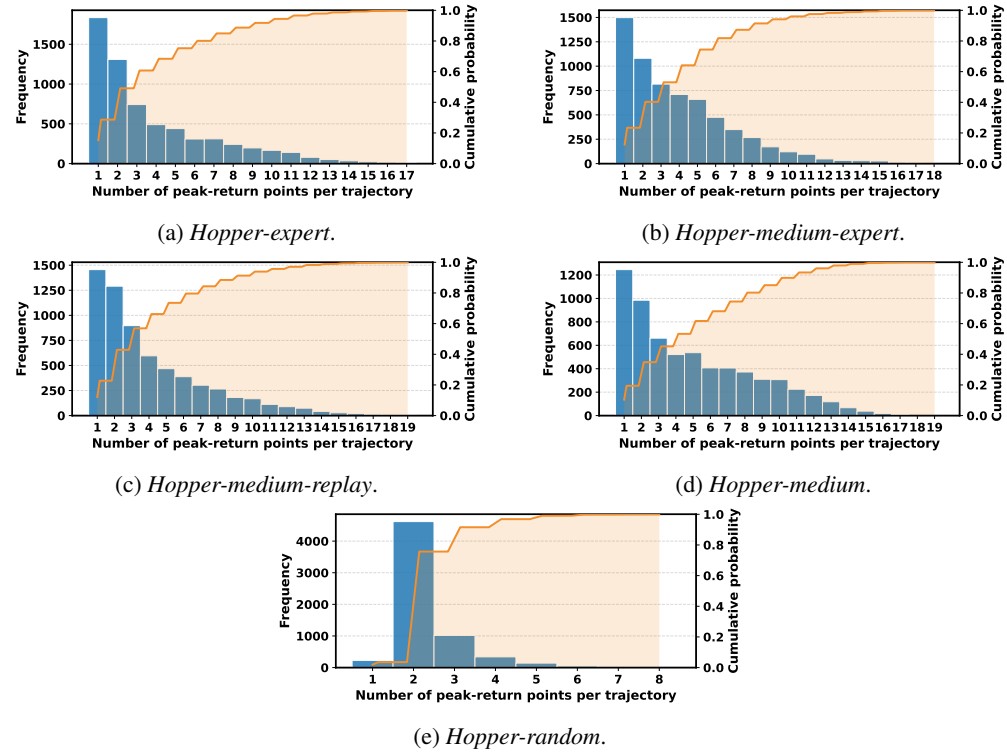

(a) *Hopper-expert.*

(b) *Hopper-medium-expert.*

(c) *Hopper-medium-replay.*

(d) *Hopper-medium.*

(e) *Hopper-random.*

Figure 15: Distribution of peak-return points per trajectory across different *hopper* datasets. The number of peak-return points varies significantly.

Similarly, we also analyzed the number of peak-return points. As shown in Figure 15, the number of peak-return points per trajectory varies significantly across different Hopper datasets. For the *hopper-expert* and *hopper-medium-expert* datasets, most trajectories have only a few peak-return points, typically around 1 or 2, indicating that in these high-quality expert datasets, the return values tend to peak at relatively fewer points during the trajectory. In contrast, the *hopper-medium-replay* and *hopper-medium* datasets exhibit a greater number of peak-return points, with *hopper-medium-replay* showing a more diverse distribution of peak points across trajectories. Finally, in the *hopper-random* dataset, almost all trajectories have peak-return points concentrated at 1 or 2, with most trajectories peaking early, reflecting the relatively low quality of this dataset.

These results suggest that the quality and complexity of the dataset significantly affect the number and distribution of peak-return points in trajectories. High-quality datasets like Hopper-expert tend to generate peaks at fewer points, while lower-quality datasets like Hopper-random show more frequent early peaks.

## C.4 ADDITIONAL RESULTS

As shown in Table 7, we also assess PRGS on narrow-coverage datasets such as random and expert. Since these datasets provide limited actionable coverage, with random datasets containing very few high-quality segments and expert datasets containing almost no low-quality segments to remove, the potential benefits of slicing are naturally constrained. Nevertheless, PRGS still achieves small but consistent improvements under these restricted conditions, indicating its ability to adapt to different levels of dataset coverage.

Table 7: Performance comparison on narrow-coverage Gym tasks.

| Task | BC | BC-PRGS | DT | DT-PRGS | PDiT | PDiT-PRGS |
|------|------|---------|------|---------|------|-----------|
| halfcheetah-random | −2.4±0.0 | −2.4±0.0 | −2.4±0.0 | −2.4±0.0 | −2.4±0.0 | −2.4±0.0 |
| hopper-random | 5.5±2.6 | 9.3±0.5 | 6.7±3.0 | 8.1±1.3 | 7.6±0.6 | 7.6±0.8 |
| walker2d-random | 1.1±0.3 | 5.9±1.8 | 5.5±0.2 | 6.4±0.7 | 6.4±0.8 | 6.3±0.6 |
| halfcheetah-expert | 92.4±0.5 | 93.2±0.1 | 91.9±0.1 | 93.6±0.3 | 90.4±1.1 | 93.2±1.0 |
| hopper-expert | 111.6±0.2 | 111.8±0.1 | 111.9±0.3 | 111.8±0.4 | 111.6±0.2 | 111.5±0.4 |
| walker2d-expert | 109.3±0.3 | 109.2±0.2 | 109.4±0.0 | 109.2±0.1 | 110.3±0.8 | 109.1±1.1 |
| **Average** | 52.9 | **54.5** ↑1.6 | 53.8 | **54.5** ↑0.7 | 54.0 | **54.2** ↑0.2 |

To further examine the consistency between PRGS and actual task achievement, we additionally report a success metric for *Maze2D*, defined as the number of times the agent enters a target-near region within a specified distance at any frame. This metric measures the success count within few episodes rather than a success rate, and a single episode may contain multiple successes. As shown in Table 8, this success measure closely mirrors the return-based improvements across all methods, indicating that PRGS does not introduce ambiguity even when subtrajectories originate from failed or partially successful trajectories. Notably, PRGS selects subtrajectories based on optimistic return rather than instantaneous rewards; since return reflects the expected cumulative outcome toward the end of the episode, it naturally aligns with this success-count metric, which is consistent with our empirical observations.

Table 8: Success count comparison within few episodes on *Maze2D* tasks.

| Task | BC | BC-PRGS | DT | DT-PRGS | PDiT | PDiT-PRGS |
|------|------|---------|------|---------|------|-----------|
| maze2d-umaze | 36.7±3.8 | 57.9±8.3 | 76.8±8.3 | 92.1±4.8 | 83.5±9.4 | 90.8±7.2 |
| maze2d-medium | 48.8±11.1 | 45.1±3.2 | 81.6±5.3 | 188.4±6.9 | 97.0±2.7 | 166.5±11.6 |
| maze2d-large | 20.1±14.5 | 24.8±2.0 | 64.2±6.4 | 221.3±16.7 | 87.2±4.9 | 183.9±10.1 |
| **Average** | 35.2 | **42.6** ↑7.4 | 74.2 | **167.3** ↑93.1 | 89.2 | **147.1** ↑57.9 |

## C.5  FAILURE MODE ANALYSIS: LIMITED TRAJECTORY COVERAGE

In addition to the main experiments, we analyze a key failure mode of PRGS that arises when the dataset lacks sufficient actionable structure. As shown in Table 7, we evaluate PRGS on **narrow-coverage datasets** such as *random* and *expert* settings, particularly in expert datasets. These datasets expose PRGS to two extreme conditions:

**Random datasets**: Only a few isolated high-return segments exist, and most trajectories contain no meaningful long-horizon structure.

**Expert datasets**: Trajectories are already near-optimal, and contain almost no low-quality segments.

Under both conditions, the potential benefit of slicing is naturally constrained. Since PRGS relies on identifying high-value transitions and separating them from low-quality ones, limited structure reduces the number of actionable peak-return points and, consequently, the effectiveness of the slicing procedure.

In these narrow-coverage regimes, PRGS behaves similarly to baseline methods: the lack of usable structure prevents the construction of meaningful composite trajectories, and the method is unable to perform effective stitching.

## D  COMPUTATIONAL OVERHEAD

As shown in Table 9, the computational overhead of PRGS comes primarily from the MMD-based return estimator, whose matrix operations introduce moderate additional memory usage during training. The greedy slicing procedure also slightly reduces training speed, although it is applied only during training and has no effect on inference. In our profiling, PRGS lowers training throughput by about 16% and increases memory usage by roughly 89%, while maintaining nearly identical inference cost. These overheads are acceptable given the consistent performance improvement brought by PRGS, and they mainly reflect the finer-grained value estimation and data selection that underpin its effectiveness.

Table 9: Computational overhead of PRGS during training and evaluation.

| Method | Training Throughput ($\uparrow$) | Memory ($\downarrow$) | Relative Gain ($\uparrow$) |
|---|---|---|---|
| DT | $1\times$ | $1\times$ | $1\times$ |
| **PRGS (training)** | $0.84\times$ | $1.89\times$ | **1.15**$\times$ |
| **PRGS (evaluation)** | $0.99\times$ | $1.08\times$ | |

## E  LLM USAGE

Large Language Models (LLMs) were used only as an assistive tool for grammar and language polishing of the manuscript. They did not contribute to research ideation, methodology design, experimental implementation, analysis, or conclusions.

