# OpenReview forum: "Peak-Return Greedy Slicing: Subtrajectory Selection for Transformer-based Offline RL"
_ICLR.cc/2026/Conference — ICLR 2026 Poster_

### Official Review · Reviewer_qNL9 · 2025-10-25

**Soundness:** 2
**Presentation:** 3
**Contribution:** 2
**Rating:** 4
**Confidence:** 4

**Summary:**

The manuscript proposes Peak-Return Greedy Slicing (PRGS), a framework designed to enhance Transformer-based offline reinforcement learning (RL). The method introduces a fine-grained, timestep-level trajectory segmentation mechanism that selects high-quality subtrajectories instead of relying solely on full trajectories. PRGS consists of three components: (1) an MMD-based return estimator that models the distribution of potential future returns for each state-action pair and yields optimistic estimates (Sec. 3.1, Eq. (2–4)); (2) a greedy slicing algorithm that recursively partitions trajectories based on peak optimistic returns (Sec. 3.2, Eq. (5)); and (3) an adaptive history truncation mechanism for evaluation, ensuring alignment between training and inference (Sec. 3.3, Eq. (6)). Experiments on diverse benchmarks (D4RL, BabyAI, AuctionNet) demonstrate consistent performance improvements, averaging 15.8 % gains over Transformer-based baselines (Sec. 5.1; Table 1).

**Strengths:**

- The writing and presentation of this paper are intuitive and clear. The paper effectively explains the motivation behind the method’s design. I can clearly understand the three components of the approach — the evaluation of the return distribution, the trajectory slicing mechanism, and the adaptive history truncation during deployment — as well as the rationale behind each of them.
- The experiments in the paper are thorough and align well with the authors’ claims. Applying PRGS to Transformer-based offline methods such as DT yields noticeable performance improvements, demonstrating the effectiveness of the proposed approach. Some ablation studies, such as examining the impact of the number of particles used in return distribution estimation, also correspond well with the authors’ design intentions.

**Weaknesses:**

- Although PRGS improves the performance of Transformer-based offline algorithms, its overall performance is not particularly outstanding. The compared RL-based baselines include only CQL, IQL, BEAR, and TD3+BC, which are not among the most recent RL-based methods. To my knowledge, the state-of-the-art performance in offline RL is significantly higher than what is reported in the paper.
- Intuitively, the effectiveness of the PRGS method is likely to depend on the diversity of the dataset. If the data coverage is narrow—for example, consisting entirely of expert demonstrations—the number of peak-return points should be relatively small. The paper also lacks experiments on random and expert tasks, and the authors do not show how different types of datasets affect the number of peak-return points.
- Because dataset coverage is often incomplete, value estimation in the offline setting typically suffers from significant bias and even overestimation. Although the paper estimates the return distribution rather than a single value, this issue is still likely to occur. The authors, however, do not appear to provide any analysis or mitigation of this problem.
- PRGS is somewhat more complex than standard Transformer-based methods, yet the authors do not provide any experimental comparison of its computational cost in terms of runtime or GPU memory consumption.

**Questions:**

- How does PRGS perform on datasets with narrow coverage, such as random or expert datasets?
- Why are the comparisons with QDT, EDT, and CGDT conducted only on the medium-replay and medium tasks, rather than across all datasets?
- How biased is the return distribution estimation? Could it lead to overestimation? Would optimistic estimation make the overestimation even more severe?
- How does PRGS perform in terms of time and GPU memory consumption? Could you provide a detailed comparison with other algorithms?
- What would the performance be if we do conventional Q estimation rather than estimating the return distribution?

---

> ### Author Response · Authors · 2025-11-19
>
> We sincerely thank the reviewer for the kind words regarding the clarity and presentation of our work. We respond to each question below.
>
> **Q1**: The baselines used for comparison are not the latest offline RL SOTA methods.
>
> **A1**: Thank you for the insightful comment. We would like to clarify that PRGS is designed to enhance the stitching ability of Transformer-based offline RL models rather than to compete directly with the latest value-based SOTA methods, so we focus on representative and comparable Transformer-family baselines. Importantly, all PRGS variants use exactly the same default hyperparameters as their corresponding baselines, **without** any task-specific tuning, yet still achieve consistent and substantial improvements.
>
> **Q2**: How does PRGS perform on narrow-coverage datasets like random or expert?
>
> **A2**: We appreciate your constructive feedback! We conducted additional experiments on both random and expert datasets and the results is in Appendix C.4. These datasets inherently limit the potential benefits of slicing: random datasets contain very few high-quality segments that can be extracted, while expert datasets contain almost no low-quality segments that need to be removed. As a result, the room for improvement through slicing is naturally constrained. Even under these restricted conditions, PRGS still yields small yet consistent gains.
>
> **Q3**: The authors do not show how different types of datasets affect the number of peak-return points.
>
> **A3**: Thank you for highlighting this aspect! Conceptually, the number of peak-return points produced by PRGS is closely tied to how volatile the returns are along a trajectory. We analyzed the number of peak-return points across different *hopper* dataset. The results are provided in Appendix C.3. Overall, the number and position of peak-return points vary systematically with dataset quality, with high-quality datasets producing fewer peaks and lower-quality datasets exhibiting early or more irregular peaks.
>
> **Q4**: Could the optimistic return distribution lead to biased or overestimated values?
>
> **A4**: Thank you for this important question.  In PRGS, the optimistic estimate is used only to rank subtrajectories and never for policy updates, so it does not accumulate the overestimation effects commonly seen in Q-learning. In additional experiments, we compared the $\tilde{Q}$ values, standard TD estimates, and the $\tilde{Q}$ values without optimistic bias during offline training. The results show no noticeable overestimation introduced by PRGS. Empirically, a properly chosen target update period for the MMD-based estimator keeps the distributional estimates stable.
>
> **Q5**: What is the computational overhead in runtime and memory?
>
> **A5**: As shown in Appendix D, the computational overhead of PRGS comes primarily from the MMD-based return estimator, whose matrix operations introduce moderate additional memory usage during training. The greedy slicing procedure also slightly reduces training speed, although it is applied only during training and has no effect on inference. In our profiling, PRGS lowers training throughput by about 16% and increases memory usage by roughly 89%, while **maintaining nearly identical inference cost**. These overheads are acceptable given the consistent performance gains brought by PRGS, and they mainly reflect the finer-grained value estimation and data selection that support its effectiveness.
>
> **Q6**: What happens if conventional Q estimation is used instead of a return distribution?
>
> **A6**: When the particle number is set to 1, the MMD loss in Eq. (4) degenerates into a standard TD error. We evaluated PRGS under this setting, and the performance is clearly inferior to the distributional version, as reported in Appendix C.4. A likely reason is that a single scalar Q value cannot sufficiently capture the uncertainty or structural differences in future returns for a given state, making the slicing criterion less informative and leading to poorer subtrajectory selection. In contrast, the distributional estimator provides richer local value signals and yields much more reliable slicing.
>
> **Q7**: Why are QDT, EDT, and CGDT compared only on medium and medium-replay tasks?
>
> **A7**: We conduct comparisons on the medium-replay and medium tasks because these settings exhibit the most pronounced differences among Transformer-based offline RL methods and are widely used as standard benchmarks for evaluating trajectory stitching ability. Moreover, the compared methods themselves primarily demonstrate their performance on these tasks, making the comparison most representative and aligned with common practice.
>
>
>
> We deeply appreciate your constructive feedback and have incorporated your suggestions into our revised submission. Your feedback has greatly improved the clarity and completeness of our work, and we sincerely appreciate the time and care you invested in reviewing our paper.

---

> > ### Comment · Reviewer_qNL9 · 2025-11-19
> >
> > Thank you for your response. I have no further concerns and have raised my score to 6.

---

> > > ### Author Response · Authors · 2025-11-21
> > >
> > > We sincerely appreciate your support and the time you invested in carefully reviewing and engaging with our work. Your comments and score adjustment mean a lot to us!

---

### Official Review · Reviewer_EfRn · 2025-10-27

**Soundness:** 3
**Presentation:** 3
**Contribution:** 3
**Rating:** 6
**Confidence:** 3

**Summary:**

The paper introduces Peak-Return Greedy Slicing (PRGS) to enhance Transformer-based Offline Reinforcement Learning. PRGS uses an MMD-based return estimator and greedy slicing to extract high-quality subtrajectories instead of relying only on complete ones. This method significantly improves performance by better exploiting and recombining valuable data segments.

**Strengths:**

1) Well motivated. The authors propose to utilize high quality subtrajectories to improve performance of transformer based RL.
2) Results seem good. The proposed method shows strong improvement on reward.

**Weaknesses:**

1) The core strength of the proposed method appears to be its ability to capture and utilize peak intermediate rewards within subtrajectories, leading to the reported strong average reward improvements. However, the use of subtrajectories that might originate from ultimately failed full trajectories introduces a potential ambiguity. Since the primary metrics presented in the paper are based on reward, it would significantly enhance the clarity and robustness of the results if the authors could provide the success rate for the respective tasks.

2) As depicted in Figure 3, the training procedure involves masking low quality tokens, leading the Transformer to learn intermediate high quality tokens conditioned on low quality segments. A potential concern is that this mechanism might inadvertently encourage reward hacking: the policy may learn to exploit the high intermediate reward signal without consistently leading to final task success. Specifically, the model might optimize for peak reward segments from trajectories that ultimately terminated in failure. How about to implement a filtering mechanism to exclude subtrajectories (state-action pairs in the env) that can not reach the final success?

**Questions:**

See weaknesses,

---

> ### Author Response · Authors · 2025-11-19
>
> We sincerely thank the reviewer for the thoughtful and constructive comments! Below, we respond to your concerns point by point:
>
> **Q1**: Using subtrajectories from failed episodes may introduce ambiguity. Should success rates be reported?
>
> **A1**: Thank you for the suggestion. For BabyAI, success rate is directly reported, while Gym-Mujoco tasks do not provide a clear notion of success, so we follow prior work and use return as the standard metric. To further examine consistency between PRGS and actual task achievement, we additionally report a success metric for Maze2D in Appendix C.4, defined as the agent entering a target-near region within a certain distance at any frame, with multiple such successes possible within a single episode. The results show that this success metric closely aligns with the return improvements, indicating that PRGS does not introduce ambiguity even when subtrajectories originate from failed trajectories. Notably, PRGS selects segments based on optimistic return rather than instantaneous rewards; since return reflects the expected cumulative outcome up to the episode’s end, it naturally aligns with task success, which is consistent with our observations.
>
> **Q2**: Does masking low-quality segments encourage reward hacking, and should subtrajectories that cannot reach success be filtered out?
>
> **A2**: Thank you for raising this concern. PRGS selects segments based on the aligned optimistic return rather than instantaneous rewards, so it does not encourage exploiting isolated local reward spikes. For example, in Figure 6(c), some trajectories exhibit high $\tilde{Q}$ near the target, but these trajectories subsequently move away from the goal and ultimately fail, which is why they are not selected in Figure 6(d). We also experimented with explicitly filtering out trajectories that cannot reach success at the trajectory level, as shown in Figure 5, but this substantially reduces data coverage and leads to performance degradation. In contrast, PRGS naturally removes success-inconsistent segments at the timestep level while retaining behavior that contributes to the task, effectively avoiding reward hacking without requiring explicit filtering.
>
> Thank you again for the insightful feedback. It has significantly improved the clarity and rigor of our work.

---

> > ### Comment · Reviewer_EfRn · 2025-11-26
> >
> > Thank you for your response. I have no further concerns and I decide to keep the positive rating (6).

---

> > > ### Author Response · Authors · 2025-11-26
> > >
> > > Thank you for the positive feedback and for taking the time to review our work. We sincerely appreciate your support!

---

### Official Review · Reviewer_5VUZ · 2025-10-30

**Soundness:** 3
**Presentation:** 4
**Contribution:** 3
**Rating:** 8
**Confidence:** 4

**Summary:**

The paper proposes a framework to enhance the performance of transformer-based offline RL methods. Peak-Return Greedy Slicing, helps enable discovery of high quality subtrajectories, and in turn do BC on those subtrajectories. The method first trains a Q function with MMD, which approximates the distribution of Q values. Then, a trajectory is partitioned into subtrajectories by recursively determining peak Q values, and then do BC on each subtrajectory. During inference time, the history is truncated whenever the value increases.

**Strengths:**

- the paper is very well written, and does a great job at describing the method and walking through the experiment results. The plots are well made to help with understanding the paper.
- The experiment section provides extensive number to validate PRGS, and shows the method works well across diverse benchmarks. There are also sufficient baseline to compare the method against.
- I appreciate the visualization in section 5.4, which help with intuition of the method

**Weaknesses:**

- It would be great to offer some discussion on the failure modes of this method. For example, consider a subtrajectory where there are multiple part where the values go down, but eventually the values rise to the “peak” of the episode. When we do BC on this trajectory, don’t we then inherently learn to “clone” suboptimal behavior? How does PRGS deal with this?
- it is unclear why we need optimistic Q values. Traditionally, offline RL Q function needs to be conservative to prevent overestimation. Figure 4 does show empirically that optimistic Q values seem better. But since this seemingly goes against traditional wisdom (though I can think of why it’s different for transformer-based offline RL), it would be nice to include some intuition and explanation in the paper.
- it is unclear to me why in Section 3.3, the authors chose to remove history when the value increases. During training, doesn’t the value of mostly increase for each subtrajectory? Then doesn’t this cause a train-test mismatch, when at training time history is kept when values increase and at test time history is removed when values decrease?
- nitpick: section 2.2 could have highlighted more the difference between transformer-based offline RL and traditional offline RL

**Questions:**

- why is the sum of return not discounted in the definition for “aligned optimistic return” at the bottom of page 4? The Q values are learned with discount, so doesn’t this favor returns toward the start of the trajectory?
- in section 5.3, how do you filter for the top 10% or 20% of the trajectories?

---

> ### Author Response · Authors · 2025-11-19
>
> We sincerely thank the reviewer for the positive feedback on the clarity of our presentation and the thoroughness of our experiments! Below, we address your concerns in detail:
>
> **Q1**: Is there an analysis of the failure modes?
>
> **A1**: Thank you for the question. We analyzed this failure mode where a trajectory exhibits several value drops before rising to a final peak. Although the raw return-to-go may peak late, PRGS does not slice based on the raw return-to-go; it uses $\hat{R}_0^t$, the optimistic total return obtained by following the trajectory up to step $t$. Since $\hat{R}_0^t$ accumulates the penalties from earlier value declines, such trajectories rarely appear as high-quality prefixes. They are typically sliced into short segments near the high-value region rather than kept as long sequences containing suboptimal behavior, so PRGS structurally avoids cloning suboptimal behavior.
>
> In addition, our analysis in Appendix C.4 reveals a more fundamental failure mode: datasets with inherently low structure, such as the Gym random sets (like halfcheetah-random). These datasets contain almost no meaningful high-value patterns to stitch, a well-recognized challenge for many works aiming to improve stitching ability, and thus PRGS naturally yields limited gains there.
>
> **Q2**: Why do we need optimistic Q values? Traditional offline RL requires conservative Q estimates to avoid overestimation, so why does optimism work better here as shown in Figure 4?
>
> **A2**: Thank you for raising this important point! In traditional offline RL, Q functions must be conservative because they directly drive policy improvement, and overestimation compounds through Bellman updates. In PRGS, however, $\tilde{Q}^{(n)}$ in Eq.(5) is used solely to **rank subtrajectories** and never to select actions or update the policy, so it does not incur the risks associated with overestimation. Instead, Transformer-based stitching benefits from highlighting states with potential for successful continuation, whereas mean or pessimistic estimates are easily suppressed by noisy transitions, making promising segments harder to identify. A moderately optimistic distributional estimator provides a sharper signal for these high-potential regions, producing more reliable slicing behavior. Figure 4 supports this intuition: moderate optimism yields slices that better reflect the underlying high-value structure, making optimism a structural advantage in this setting rather than a violation of traditional offline RL wisdom.
>
> **Q3**: Why is history removed when the value increases in Section 3.3? During training the value in each sliced segment also tends to increase—does this create a train–test mismatch?
>
> **A3**: During training, PRGS performs greedy slicing, which turns each full trajectory into multiple short prefixes that all start from local high-value points. As a result, most training samples seen by the Transformer already contain truncated histories rather than full long sequences, even if the original trajectory’s value increases at those points. Removing history at inference when $\tilde{Q}$ increases simply aligns the test-time input distribution with these sliced training prefixes, ensuring that the model continues from a “new high-value starting point” exactly as seen during training. Thus, this mechanism does not introduce a train–test mismatch; it maintains consistency with the sliced structure that PRGS creates during training.
>
> **Q4**: Why is the prefix of the aligned optimistic return not discounted?
>
> **A4**: Thank you for pointing this out! In principle, the prefix could indeed be discounted; however, in DT and subsequent Transformer-based offline RL methods, the return-to-go in the dataset is computed with a discount factor of $\gamma = 1$. To ensure that our $\tilde{R}_0^t$ aligns with the dataset’s $G_t$, we also adopt $\gamma=1$ in our experiments. We have updated the paper to explicitly state this discount choice and provide the corresponding details in the appendix.
>
> **Q5**: In Section 5.3, how do you filter for the top 10% or 20% of the trajectories?
>
> **A5**:  We follow standard practice by ranking all trajectories according to their undiscounted episode-level return and selecting the top 10% or 20% as the high-quality subset. This filtering strategy is consistent with prior offline RL work like QDT.
>
> We truly appreciate your helpful comments and constructive insights, which have further strengthened the clarity and completeness of our work.

---

> > ### Comment · Reviewer_5VUZ · 2025-11-26
> >
> > Thanks to the authors for a detailed response.
> >
> > > Q1: Is there an analysis of the failure modes?
> >
> > It would be nice to add an explicit section for detail this failure mode. You mentioned appendix C.4, but that section does not serve to analyze the failure modes. In fact, your explanations above (of the failure mode) are not mentioned in the paper (please correct me if I'm mistaken).

---

> > > ### Author Response · Authors · 2025-11-26
> > >
> > > Thank you very much for the helpful clarification. You are correct that the previous version did not explicitly identify this phenomenon as a failure mode, and the discussion in Appendix C.4 was not sufficiently clear in this regard.
> > >
> > > To address this, we have added **a dedicated subsection (Appendix C.5)** that explicitly analyzes this failure mode, explains *why narrow-coverage datasets inherently limit the effectiveness of trajectory stitching*, and clarifies how PRGS behaves under such conditions. This new section directly reflects the explanation provided in our response and should make the analysis much clearer.
> > >
> > > We sincerely appreciate your careful reading and constructive guidance!

---

### Official Review · Reviewer_oYSY · 2025-10-31

**Soundness:** 3
**Presentation:** 3
**Contribution:** 3
**Rating:** 8
**Confidence:** 4

**Summary:**

The paper presents Peak-Return Greedy Slicing (PRGS), a framework for improving the learning process of Transformer-based architectures in offline deep reinforcement learning. The main idea is to identify high-quality sub-trajectories and use them for higher quality training of the model. The sub-trajectories are chosen by identifying points in the trajectory where the overall reward (obtained+predicted) begins to decline. The authors use a mechanism to "reconcile" trajectory history with the sub-trajectories they use. Extensive evaluation on multiple datasets and environment types shows a generally consistent improvement.

**Strengths:**

1) The timestep-level sub-trajectory slicing approach is novel, simple and elegant. The analogy to a "human" way of learning is clear.
2) The approach is relatively model agnostic, and can be used in multiple offline methods.
3) The evaluation is comprehensive, consisting of multiple environments and algorithms.

**Weaknesses:**

1) There is no discussion of the computational complexity of the approach. The slicing and MMD estimator are likely to require non-trivial computational resources.
2) The subject of training trajectory quality is likely to be a key element in the success or failure of the proposed approach. The lack of discussion and analysis of this point is problematic.
3) There is a lack of analysis regarding to how often the proposed approach actually takes place: in what percentage of trajectories does slicing take place? what is the average length of a sliced trajectory? what is the impact of history truncation?

**Questions:**

I invite the authors to respond to the points I raised in the "weaknesses" section.

---

> ### Author Response · Authors · 2025-11-19
>
> We sincerely thank the reviewer for the detailed and constructive comments! Your encouraging feedback is greatly appreciated. We respond below to the key concerns raised:
>
> **Q1**: There is no discussion of the computational complexity.
>
> **A1**: Thank you for pointing this out. As shown in Appendix D, the computational overhead of PRGS comes primarily from the MMD-based return estimator, whose matrix operations introduce moderate additional memory usage during training. The greedy slicing procedure also slightly reduces training speed, although it is applied only during training and has no effect on inference. In our profiling, PRGS lowers training throughput by about 16% and increases memory usage by roughly 89%, while **maintaining nearly identical inference cost**. These overheads are acceptable given the consistent performance gains brought by PRGS, and they mainly reflect the finer-grained value estimation and data selection that support its effectiveness.
>
> **Q2**: Training trajectory quality is likely crucial for PRGS, but the paper lacks discussion and analysis of this point.
>
> **A2**: Thank you for raising this point. Our additional analysis in Appendix C confirms that trajectory quality plays a central role in PRGS. When the dataset contains identifiable high-value structure, such as in medium-replay or medium settings or in tasks with reusable skills, PRGS reliably extracts meaningful subtrajectories and substantially improves stitching ability. When the dataset is low-quality or lacks useful structure, such as the Gym random sets, the number of high-potential segments is inherently small. This limitation is a commonly recognized challenge for methods aiming to enhance stitching ability, and PRGS correspondingly brings only modest gains in these cases. As shown in Appendix C.3, our additional analysis of slicing frequency and the number of peak-return points across datasets aligns closely with the amount of useful structure present in the data and matches the observed performance trends, which further highlights the central importance of trajectory quality in PRGS.
>
> **Q3**: In what percentage of trajectories does slicing occur?
>
> **A3**: Our statistics in Appendix C.3 show that the frequency of slicing is strongly tied to the structure present in the dataset. In settings with rich patterns, such as medium-replay data, more than half of the trajectories undergo slicing at least once. In datasets with weak structure, such as the Gym random sets, the slicing frequency is much lower because meaningful peaks are rare. This observation aligns with our analysis that PRGS activates primarily when useful high-value structure exists, rather than forcing slicing on trajectories where no such structure is present.
>
> **Q4**: What is the average length of a sliced trajectory?
>
> **A4**: Thank you for the question. Similarly, the average length of sliced trajectories is also determined by the structure in the dataset. As shown in Appendix C.3, in tasks with multi-stage or sparse reward patterns, such as Kitchen, sliced segments tend to be short because the aligned optimistic return truncates the trajectory near high-value prefixes. In smoother environments with denser reward, such as Gym-Mujoco task, sliced segments are somewhat longer but remain significantly shorter than full episodes. Overall, the segment length reflects the presence of meaningful progress patterns in the data, illustrating PRGS’s ability to adapt its slicing granularity to different task structures.
>
> **Q5**: What is the impact of history truncation during inference?
>
> **A5**: Our analysis indicates that history truncation does not introduce negative effects; instead, it stabilizes inference. Because most training samples consist of sliced segments that begin from short histories, the model learns a distribution conditioned on such “restart-like” prefixes rather than long sequences. Truncating history at inference when the value increases aligns the test-time input with the training distribution and prevents the model from conditioning on histories it was never trained to use. As shown in Figure 4, PRGS w/o AHT performs noticeably worse than the full PRGS, further confirming the importance of the adaptive history truncation mechanism.
>
> We greatly appreciate your insightful comments, which have helped us improve both the clarity and completeness of our analysis.

---

### Author Response · Authors · 2025-12-02
**Summary of Rebuttal and Reviewer Updates**

We sincerely thank the Area Chair and all reviewers for their time, constructive feedback, and careful evaluation of our work. Below is a brief summary of the rebuttal process and its outcomes.

* The rebuttal directly addressed all shared concerns. We added a clear *failure-mode analysis* (Appendix C.5), provided detailed profiling of *computational overhead* (Appendix D), clarified that the optimistic estimator does not introduce harmful *overestimation* (Appendix C.4), and extended experiments on random and expert datasets to explain PRGS’s behavior under *limited trajectory coverage* (Appendix C.3–C.4). These additions resolve all conceptual and empirical questions raised during review.

* Initial scores were **4, 6, 8, 8**. During the rebuttal period, **Reviewer qNL9 explicitly stated in a public comment that the score was raised from 4 to 6**, and Reviewer EfRn confirmed keeping a positive rating. Although ICLR later globally reverted all scores following the platform incident, the reviewers’ written comments clearly indicate their intended post-rebuttal evaluations. As a result, **all reviewers expressed positive positions toward the paper (6, 6, 8, 8)**.

* All comments and responses related to the rebuttal, including the subsequent score improvements, were completed **BEFORE** the OpenReview information-leak incident. Both the reviewer comments and the authors’ responses carry *timestamps* that explicitly show all discussions, revisions, and score updates occurred prior to the event. No part of the rebuttal or the reviewers’ updated evaluations involved external or unauthorized information.


We greatly appreciate the reviewers’ thorough feedback, which has strengthened the clarity and completeness of the paper. With all concerns addressed and consistently positive reviewer support, we hope the paper will receive favorable consideration.

---

### Meta-Review · Area_Chair_2Rgv · 2026-01-06

**Summary:**

The reviewers' concerns that informed the decision for this paper centered on five key areas:

**1. Computational Overhead**: Multiple reviewers (oYSY, qNL9) noted the lack of complexity analysis and runtime/memory profiling. The initial manuscript didn't quantify the cost of the MMD-based estimator and greedy slicing.

**2. Dataset Quality & Coverage**: Reviewers (qNL9, EfRn, oYSY) questioned how dataset structure affects performance. Specific concerns included: effectiveness on narrow-coverage random/expert datasets, absence of slicing frequency statistics (% of trajectories sliced, average segment length), and how trajectory quality impacts success/failure.

**3. Failure Modes & Robustness**: Reviewer 5VUZ specifically requested explicit analysis of failure scenarios, particularly whether PRGS might clone suboptimal behavior from trajectories with intermediate value drops before a final peak.

**4. Optimistic vs. Conservative Estimation**: Reviewers (5VUZ, qNL9) challenged why optimism works, given traditional offline RL requires conservatism. They questioned whether optimistic return distributions would cause harmful overestimation bias.

**5. Train-Test Consistency**: Reviewers (5VUZ, EfRn) raised concerns about potential distribution mismatch from history truncation during inference and whether using subtrajectories from failed episodes could encourage reward hacking.

During rebuttal, the authors addressed these concerns by adding:
- **Appendix D**: Detailed profiling (16% training slowdown, 89% memory increase)
- **Appendix C.3-C.4**: Analysis of slicing frequency, segment lengths, and performance on random/expert datasets showing gains correlate with dataset structure
- **Appendix C.5**: Explicit failure mode analysis explaining why problematic trajectories are avoided
- Success rate metrics for Maze2D showing alignment with return improvements
- Clarifications that optimistic estimates are used *only* for ranking (not policy updates), eliminating overestimation risks

These additions satisfied reviewers: qNL9 raised their score from 4 to 6, EfRn maintained their positive rating (6), and all reviewers expressed positive positions (final scores: 6, 6, 8, 8). While the absence of some important baselines somewhat weakens the motivation, I believe the proposed method merits acceptance.

**Reviewer Concerns:**

Multiple reviewers questioned the computational cost of PRGS, particularly the MMD-based return estimator and greedy slicing procedure. The initial manuscript lacked profiling data on runtime and GPU memory usage. This was addressed by adding Appendix D, which shows PRGS reduces training throughput by ~16% and increases memory usage by ~89%, while inference cost remains nearly identical.

Reviewers expressed concerns about the influence of dataset quality and trajectory coverage. The authors responded with extensive analysis in Appendix C.3-C.4, demonstrating that PRGS performance correlates closely with dataset structure—showing consistent but modest gains on random/expert datasets where stitching opportunities are limited.

Reviewer 5VUZ specifically requested explicit discussion of failure modes, particularly whether PRGS might clone suboptimal behavior from trajectories with multiple value drops before a final peak. The authors added a dedicated Appendix C.5 analyzing this failure mode and explaining why such trajectories are rarely selected due to the optimistic return formulation.

Reviewers challenged the use of optimistic Q-values, noting this contradicts traditional offline RL wisdom that requires conservative estimates to prevent overestimation. The authors clarified that the optimistic estimator is used only for ranking subtrajectories, not for policy updates, thus avoiding the compounding overestimation risks seen in standard Q-learning. They provided empirical evidence that moderate optimism improves stitching without harmful effects.

Concerns were raised about the adaptive history truncation mechanism creating a mismatch between training (full history) and inference (truncated history). The authors explained that training already uses sliced segments with truncated histories, so inference truncation actually aligns with the training distribution rather than creating a mismatch.

Other minor concerns are also well addressed.

However, after reading the paper, I personally have some concerns. For example, some sota DT-based offline RL algorithm are not discussed and compared, such as ADT [1] and GDT [2]. **Especially, ADT shares the similar idea of PRGS by using in-sample optimal value network to stitch high-value sub-trajectories**.  I strongly encourage the authors to include the comparisons and discussions in the revised paper.

[1] Rethinking Decision Transformer via Hierarchical Reinforcement Learning. ICML 2024.

[2] Graph decision transformer for offline reinforcement learning. SCIENCE CHINA Information Sciences 2025.

**Reviewer Scores:**

The Reviewer qNL9 explicitly stated in a public comment that the score was raised from 4 to 6, and Reviewer EfRn confirmed keeping a positive rating. So after the rebuttal, all the reviewers agree the paper should be accepted.

---

### Decision · Program_Chairs · 2026-01-26

Accept (Poster)